# A specific microbial consortium enhances Th1 immunity, improves LCMV viral clearance but aggravates LCMV disease pathology in mice

Daphne Kolland[1], Miriam Kuhlmann[2], Gustavo P. de Almeida [2,3], Amelie Köhler[1], Anela Arifovic[1], Alexandra von Strempel[4], Mohsen Pourjam [5], Silvia Bolsega [6], Christine Wurmser [2,3], Katja Steiger [7], Marijana Basic [6], Klaus Neuhaus [5], Carsten B. Schmidt-Weber [1,8], Bärbel Stecher [4,9], Dietmar Zehn [2,3] ✉ & Caspar Ohnmacht [1] ✉

Anti-viral immunity can vary tremendously from individual to individual but mechanistic understanding is still scarce. Here, we show that a defined, low complex bacterial community (OMM[12]) but not the general absence of microbes in germ-free mice leads to a more potent immune response compared to the microbiome of specific-pathogen-free (SPF) mice after a systemic viral infection with LCMV Clone-13. Consequently, gnotobiotic mice colonized with OMM[12] have more severe LCMV-induced disease pathology but also enhance viral clearance in the intestinal tract. Mechanistically, single-cell RNA sequencing analysis of adoptively transferred virus-specific T helper cells and endogenous T helper cells in the intestinal tract reveal a stronger pro-inflammatory Th1 profile and a more vigorous expansion in OMM[12] than SPF mice. Altogether, our work highlights the causative function of the intestinal microbiome for shaping adaptive anti-viral immunity with implications for vaccination strategies and anti-cancer treatment regimens.

The intestinal microbiome is increasingly understood to influence multiple systems of its host including the hormone system, the central nerval system and even behavior. Another prominent role of the intestinal microbiome has been documented for its contribution to epithelial barrier integrity and immune maturation[1]. Animals raised in the absence of any bacteria (germ-free animals) fail to fully develop normal secondary organ structures in the intestinal tract such as Peyer's Patches and mature isolated lymphoid follicles in the lamina propria. Moreover, conserved structural features or bacterial metabolites of individual bacteria present in the intestinal microbiome can modulate the functionality of different types of immune cells[1]. There are many examples of how bacterial symbionts or pathobionts modulate intestinal T helper cells including the generation of Th17 cells or regulatory T cells (Tregs)[2–5]. By contrast, only few reports indicate a

[1]Center of Allergy and Environment (ZAUM), Technical University and Helmholtz Center, Munich, Germany. [2]Division of Animal Physiology and Immunology, School of Life Sciences Weihenstephan, Technical University of Munich, Freising, Germany. [3]Center for Infection Prevention (ZIP), School of Life Sciences Weihenstephan, Technical University of Munich, Freising, Germany. [4]Max von Pettenkofer Institute of Hygiene and Medical Microbiology, Faculty of Medicine, LMU, Munich, Germany. [5]Core Facility Microbiome ZIEL – Institute for Food & Health, Technical University of Munich, Freising, Germany. [6]Institute for Laboratory Animal Science and Central Animal Facility, Hannover Medical School, Hannover, Germany. [7]Institute of Pathology, School of Medicine and Health, Technical University Munich, Munich, Germany. [8]Member of the German Center of Lung Research (DZL), Partner Site Munich, Munich, Germany. [9]German Center for Infection Research (DZIF), partner site LMU, Munich, Germany. ✉e-mail: dietmar.zehn@tum.de; caspar.ohnmacht@helmholtz-munich.de

microbiome-dependent induction of Th1 cells typically associated with anti-viral immunity and such microbiome effects are typically associated to parallel effects on other T helper cell lineages[6,7].

Noteworthy, the importance of elucidating microbiome effects on viral infections has manifested during the coronavirus disease (COVID-19) pandemic. Many reports indicate a correlation between the intestinal microbiome and disease severity of COVID-19[8,9]. Moreover, viral infections can cause transient dysbiosis of the intestinal microbiota as a bystander effect, as observed in the common cold and mild cases of influenza as well as SARS-CoV-2 infections[10]. In chronic viral infections like HIV or HBV the impact of the intestinal microbiota is less obvious. Nevertheless, the microbiome of HIV-infected individuals is decreased in bacterial richness and diversity, associated with systemic inflammation and immune activation[11]. Moreover, individuals with a controlled HIV replication have a more similar microbiome to HIV-uninfected individuals[12]. Altogether, these findings suggest that manipulation of the intestinal microbiome offers an additional approach to improve disease and treatment regimens but warrants more mechanistic studies to prove cause-effect relationship. The high intra-individual variation of the intestinal microbiome is also suspected to affect responsiveness to vaccination regimen[13]. Indeed, even effector functions such as cytokine secretion by cytotoxic CD8+ T cells can be modulated by individual bacterial members of the microbiome[14]. This regulation of effector function by the intestinal microbiome has additionally been observed in animals raised under a more 'dirty' environment: Laboratory animals co-housed with mice from pet stores or 'wildling' animals showed a different responsiveness to viruses and intracellular pathogens such as *Listeria monocytogenes*—a phenomenon that was attributed to microbiome effects on innate immunity[15,16].

The ability of the intestinal microbiome to affect adaptive immune responses after systemic virus infections have not been addressed in detail since the interplay between the microbiome community, invasive viruses and host interactions are highly complex and require clearly defined settings to draw effect-response conclusions. Potential mechanisms by which the microbiome can influence anti-viral immunity vary according to host cell specificity of the virus, viral infection kinetic and pathogenesis[10]. One obvious and therapeutically highly attractive way how intestinal bacteria contribute to antiviral effects may be the generation of metabolites and continuous stimulation of the host immune system by additional bacteria-derived factors[17]. Yet, such functionally active molecules are very difficult to identify—not least due to the enormous variability of individual microbiome composition among human beings and even across animal facilities housed under specific pathogen free conditions. Infection with the *Lymphocytic choriomeningitis virus* (LCMV) elicits a systemic viral infection and is widely used to study anti-viral T cell responses. While cytotoxic CD8+ T cells play a key role for disease pathology and viral clearance, this virus model has also been used to investigate CD4+ T cell help to regulate CD8+ T cell differentiation[18,19]. Early on, CD4+ T cell help has been shown to sustain the cytotoxic potential of CD8+ T cells during chronic viral infection[20–22]. Even though CD8+ cytotoxic T cells are the main cell type involved in viral clearance after LCMV infection, the role of CD4+ T helper cells is still less well understood. T helper cell differentiation is influenced by the intestinal microbiome in multiple ways but whether this occurs also for virus-specific T cells remains largely unknown.

The use of animals with a defined and minimal microbiota offers the possibility to study clear causality between the intestinal microbiome and immunity to systemic viral infection. In the present study, specific-pathogen-free (SPF), germ-free (GF) mice and mice with a defined low-complex microbiota (OMM[12])[23] are infected with the chronic LCMV Cl-13 strain. Surprisingly, OMM[12] animals show rapid body weight loss and improved viral clearance that is associated with

excessive proliferation and a cytotoxic Th1 profile of virus-specific T helper cells. Thus, the OMM[12] bacterial community enhances adaptive anti-viral immunity and disease pathology. In other settings, these findings have implications for microbiota-mediated optimization of treatment regimens in anti-cancer immunity or vaccination strategies.

## Results

### OMM[12] mice show enhanced disease pathology and viral clearance associated with stronger T helper cell response after systemic viral infection

To investigate the role of the intestinal microbiota on the immune response of a systemic viral infection, we infected either SPF animals, animals colonized with a defined low-complex bacterial consortium called OMM[12] [23] and GF animals with the LCMV Cl-13 strain (Fig. 1A). Surprisingly, OMM[12] animals showed substantial more body weight loss already at day 4 post-infection (4 dpi) compared to SPF and this difference was maintained at 8 dpi with OMM[12] animals reaching endpoint criteria (Fig. 1B). Noteworthy, uninfected OMM[12] animals have a normal body weight comparable to SPF animals at 12 weeks of age (Supplementary Fig. 1A), in line with previous reports[24,25]. GF animals also lost body weight but only to the same extent as SPF animals (Fig. 1B). Since LCMV is a non-cytopathic virus, more severe body weight loss is a sign of stronger immune activation which may result in better viral clearance[26]. Indeed, we detected a lower viral titer in the small intestine of OMM[12] compared to SPF and GF animals, which was reproduced in the spleen (Fig. 1C). Viral clearance in LCMV Cl-13 infection is largely driven by virus-specific cytotoxic CD8+ T cell response and exhaustion of these cells is causative for virus persistence[27,28]. However, we were not able to detect significant effects of the microbial status on the CD8 T cell response[29]. Therefore, we turned our focus to the CD4 T helper cell response since this cell type is another crucial player in the regulation of adaptive anti-viral immunity and is known to be affected by the microbiome[20–22]. As expected, LCMV Cl-13 infected GF and OMM[12] animals maintained a gradual reduction in the frequency of microbiome-dependent RORγt+ Tregs and Th17 cells while Gata3+ Th2 cells showed the opposite behavior (Supplementary Fig. 1B). The diminished but not ablated pTreg and Th17 population has already been shown before in OMM[12] and GF mice[30]. However, LCMV Cl-13 infection resulted in higher frequencies of T helper cells in the lamina propria of OMM[12]-colonized and GF mice with high expression levels of the transcription factor T-bet, a major regulator for Th1 effector cell differentiation (Fig. 1D, E)[31]. In absolute numbers, this difference was again only retained in OMM[12] in the small intestine and not apparent in spleen (Fig. 1E), possibly due to different levels of control by microbiome-induced RORγt+ regulatory T cells (Supplementary Fig. 1B). Noteworthy, we also detected higher frequencies of CD4+ T cell producing the Th1-associated cytokines IFNγ and TNFα in the lamina propria of OMM[12] animals after restimulation with the LCMV MHC class II-restricted peptide gp61 (Figs. 1F, G and Supplementary Fig. 1C). Both cytokines may synergistically contribute to enhanced disease pathology in OMM[12] animals, either locally in the intestinal tract or more systemically by affecting even cells of the central nervous system[32,33]. Altogether, these results suggest that OMM[12] animals show more severe body weight loss and enhanced viral clearance, associated with a stronger CD4 T cell response after systemic LCMV Cl-13 infection.

### CD4+ T helper cell depletion partially rescues the OMM[12] phenotype

As our results indicated a microbial effect of the OMM[12] bacterial consortium on the virus-specific CD4 T cell response, we next investigated whether CD4 T cells were causative for the microbiome effects on disease pathology and viral clearance. Hence, we depleted CD4 T cells prior to and during LCMV Cl-13 infection in gnotobiotic OMM[12] and SPF animals (Fig. 2A). CD4+ T cells were efficiently depleted in the

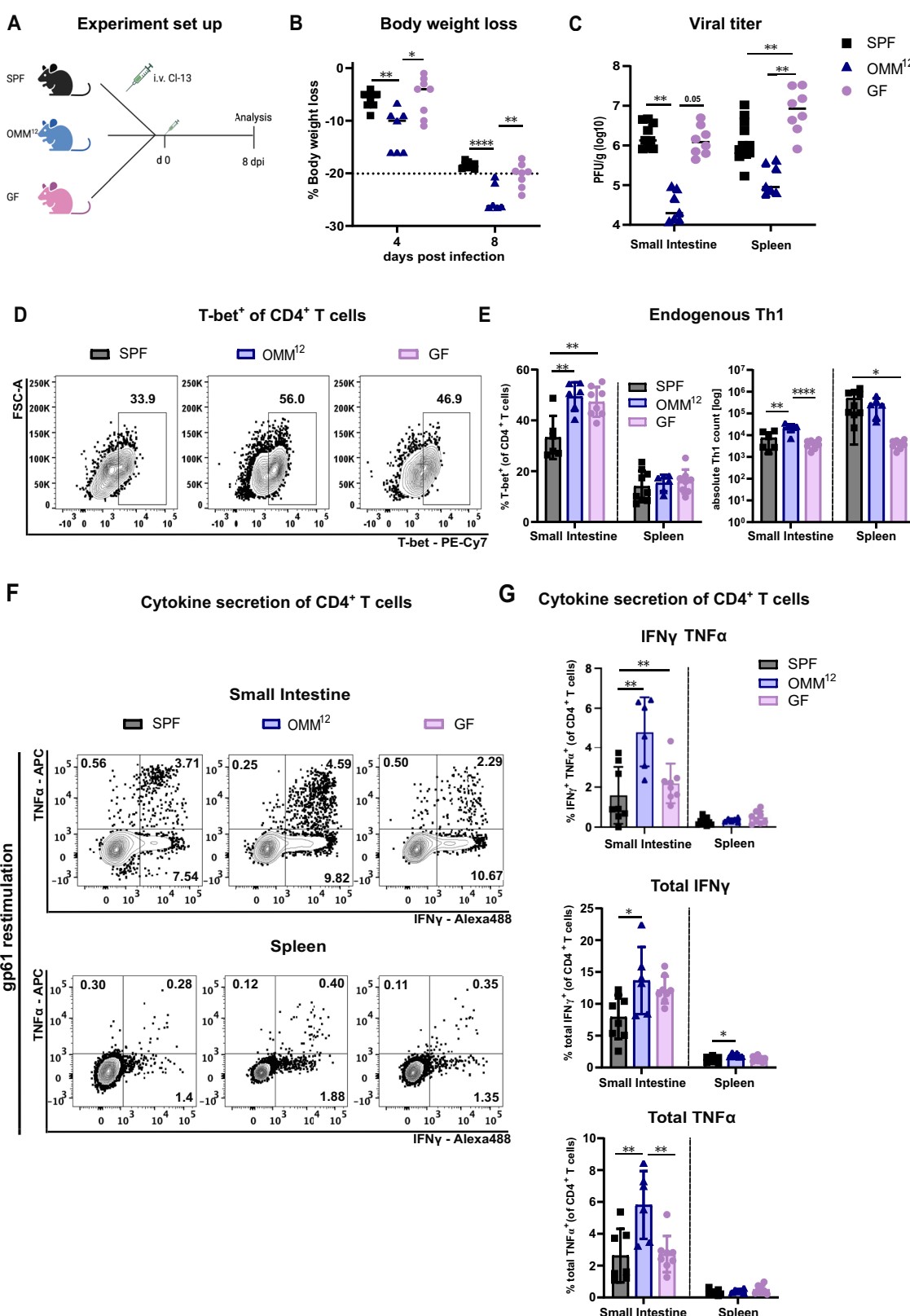

spleen at 8 dpi (Supplementary Figs. 2A, B). Former studies have shown no differences in disease pathology during the acute phase of LCMV Cl-13 in SPF mice[19,21]. Accordingly, CD4 depletion showed no effects on body weight loss in the SPF group (Fig. 2B). As CD8⁺ T cells might be affected by CD4 T cell depletion we stimulated splenocytes and lymphocytes isolated from the mesenteric lymph nodes (mLN) with the LCMV MHC class I-restricted peptide gp33 to investigate cytokine

secretion of virus-specific CD8⁺ T (CTL) cells (Supplementary Fig. 2C). CTLs did show a trend for higher secretion of IFNγ upon CD4⁺ T cell depletion; nevertheless, we could see no differences in inflammatory and cytotoxic cytokine secretion nor in other subsets of CD8⁺ T cells between OMM¹² and SPF mice regardless of treatment (Supplementary Figs. 2D, E). This result is not contradictory to other reports that showed a prominent T helper role to sustain CD8 T cell responses only

**Fig. 1 | OligOMM¹² colonized mice display a stronger body weight loss but improved viral clearance associated with an enhanced Th1 immune response.** SPF (gray), OligOMM¹²-colonized (OMM¹², blue) and germ-free (GF, pink) mice were intravenously infected with LCMV Clone-13 and analyzed at day 8 post infection (dpi). **A** Illustration of the experimental groups and setup. Created in BioRender (https://BioRender.com/q80p879). **B** Percentage of body weight loss of indicated groups at 4 and 8 dpi. Dotted line indicates 20% body weight loss (endpoint criteria). *P* values are as followed: 4 days post infection (SPF vs OMM¹² = 0.0052, OMM¹² vs GF = 0.0107) and 8 days post infection (SPF vs OMM¹² < 0.0001, OMM¹² vs GF = 0.0034). **C** Plots depict plaque forming units (PFU/g) in the small intestine and spleen of GF, SPF and OMM¹² colonized animals at 8 dpi. *P* values are as followed: small intestine (SPF vs OMM¹² = 0.0096, OMM¹² vs GF = 0.05) and spleen (SPF vs GF = 0.0061 and OMM¹² vs GF = 0.0048). **D** Representative flow cytometry plots of Th1 (T-bet^hi of CD4⁺ T cells (**E**) cell frequencies and absolute cell numbers in small intestine and spleen. *P* are as followed for small intestine (SPF vs OMM¹² = 0.0017 and SPF vs GF = 0.0032), for absolute Th1 counts in small intestine (SPF vs

OMM¹² = 0.0010 and OMM¹² vs GF < 0.0001) and in spleen (SPF vs GF = 0.0161). **F** Representative flow cytometry plots and frequencies (**G**) of cytokine⁺ CD4⁺ T cells after restimulation with the virus-peptide gp61. *P* values are as followed: IFNγTNFα small intestine (SPF vs OMM¹² = 0.0013 and SPF vs GF = 0.0071), total IFNγ small intestine (SPF vs. OMM¹² = 0.0109), spleen (SPF vs OMM¹² = 0.0213) and total TNFα small intestine (SPF vs OMM¹² = 0.0064 and OMM¹² vs GF = 0.0037). For small intestine samples of two animals of the same group were occasionally pooled to increase the yield. Data from two independent experiments are shown. SPF *n* = 9, OMM¹² *n* = 6, GF *n* = 8 mice. Each dot represents an individual mouse (or occasionally pooled cells from two individual mice of the same group) and mean ± SD from at least two independent experiments is shown. Statistical analysis for (**B**, **C**, **E**, and **G**) was performed by using one-way ANOVA with Tukey correction for multiple comparison. *P* value of <0.05 was considered statistically significant. *$p < 0.05$, **$p < 0.01$, ***$p < 0.001$, ****$p < 0.0001$. Source data are provided as a Source Data file.

during the chronic infection phase or after recall infections[18,21,22,34–37]. However, depletion of CD4 T cells rescued OMM¹² animals from more severe body weight loss at day 4 after infection (Fig. 2B). This trend could also be observed at 8 dpi. As expected, we could confirm a major difference in body weight loss in isotype treated OMM¹² compared to SPF groups (Fig. 2B). Moreover, isotype-treated OMM¹² mice showed again a lower viral titer in the small intestine compared to isotype-treated SPF animals but we could not observe any effect of CD4 T cell depletion on viral clearance (Fig. 2C). Histological examination of the small intestines of each group revealed no significant epithelial changes or mucosal architecture (Fig. 2D) nor could we detect a significant change in intestinal inflammation, epithelial apoptosis or number of intraepithelial lymphocytes (Fig. 2E–G).

Lastly, we aimed to understand whether LCMV Cl-13 infection itself affected the abundance of individual bacterial species in OMM¹² animals. Longitudinal qPCR analysis of the 12 strains after LCMV Cl-13 infection revealed only minor differences between the CD4 depleted and the isotype treated OMM¹² group early after infection which is possibly attributable to a stress response (Figs. 2H–J and Supplementary Fig. 2F–N). Throughout the course of the infection, some minor impact of LCMV Cl-13 on the OMM¹² consortium can be seen at 8 dpi. These effects include an increase in *Enterococcus faecalis*, *Akkermansia muciniphila* and *Clostridium innocuum* (Fig. 2E, H, J). Such small shifts may be caused by systemic inflammation equally affecting the gastrointestinal tract, but CD4 T cell depletion only marginally affected relative abundance of most species. Altogether, our results demonstrate that CD4 T cells contribute to the enhanced body weight loss early after infection but not viral clearance in OMM¹²-colonized animals. Yet, LCMV Cl-13 infection only marginally affected the composition of the OMM¹² consortia, and any immunomodulatory effect may thus occur early or is even present before viral infection.

## Higher expansion and effector function of LCMV-specific CD4 T cells in OMM¹² animals

The CD4 T cell depletion experiment suggested an important role for polyclonal CD4 T cells to mediate effects of the OMM¹² consortia on anti-viral immunity. To address the impact of the OMM¹² bacterial consortia on virus-specific T cells, we next adoptively transferred naïve CD45.1⁺ SMARTA T cells that harbor a transgenic TCR specific for the H2-I-A^b-restricted LCMV gp_{61-80} epitope (glycoprotein residues 61-80) into SPF, OMM¹² and GF animals before infection with LCMV Cl-13 (Fig. 3A). Again, we found a stronger weight loss in OMM¹² animals (Fig. 3B) associated with a lower viral titer (Fig. 3C). Interestingly, SMARTA T cells showed a much higher frequency and up to a 10fold higher total cell number in the lamina propria of LCMV Cl-13-infected OMM¹²-colonized animals compared to SPF and GF animals while this effect was not seen in the spleen (Fig. 3D, E). Restimulation with the MHC-II-restricted immune dominant LCMV peptide gp61 revealed

higher production of IFNγ among SMARTA T cells isolated from the lamina propria compared to those from the spleen (Fig. 3F, G). Noteworthy, the highest frequency was again observed in OMM¹²-colonized animals (Fig. 3F, G). OMM¹²-colonized animals also showed the highest frequency of TNFα-producing SMARTA T cells in both organs (Fig. 3F, G). This improved effector function could also be seen in the endogenous CD4 T cell compartment isolated from the small intestine of OMM¹² mice (Fig. 3H). The frequency of cytokine⁺ T cells in the spleen was substantially lower but followed the same trend with higher levels in OMM¹²-colonized animals (Fig. 3F–H). As the effects of the OMM¹² consortia were most pronounced in the lamina propria of the small intestine – a site in proximity with the intestinal microbiota – virus-specific T helper cells may be influenced locally by OMM¹² consortia-derived signals. In summary, these results suggest that the OMM¹² bacterial consortium allows for a stronger virus-specific T helper cell response but maintains its community composition even during LCMV infection.

Additionally, we investigated whether the exacerbated pro-inflammatory T cell response in the lamina propria correlated with a generally higher inflammation. Hence, we performed histological examination of OMM¹² and SPF animals with adoptively transferred SMARTA T cells 8 dpi (Supplementary Figs. 3A–C). No differences in inflammation score (Supplementary Fig. 3A) nor in the number of lymphocytes infiltrating the luminal intestinal epithelium or the numbers of epithelial apoptosis (Supplementary Figs. 3B, C) were observed in OMM¹² animals.

To rule out any contamination of OMM¹² animals and assess the impact of LCMV Cl-13 infection on microbial composition, we collected feces at different timepoints of the infection and performed 16S sequencing of SPF and OMM¹² mice (Figs. 3A and Supplementary Fig. 3D). The analysis confirmed the absence of contamination in the OMM¹² and some differences in bacterial composition at 8 dpi including a relative expansion of *Escheria Shigella* and *Bacteroides* at the expense of *Lactobacillus* in SPF animals (Supplementary Fig. 3D). Again, the OMM¹² consortium was only marginally affected by LCMV Cl-13 infection in line with their stable community also in other infections[23] and our qPCR results (Fig. 2H–J). Thus, OMM¹² consortia harbor such a pro-Th1 supporting effect intrinsically which can affect the strength of anti-viral immunity in the first days after infection, e.g., here measurable by increased body weight loss.

## Virus-specific CD4⁺ T helper cells show phenotypical heterogeneity in gnotobiotic animals post LCMV Cl-13 infection

Given the profound effect of the OMM¹² consortium on T helper cells, we next aimed to gain a comprehensive view of the differentiation of LCMV-specific SMARTA T cells under different gnotobiotic conditions. Thus, we again transferred congenically marked CD45.1⁺ naïve SMARTA T cells into SPF, OMM¹² and GF animals before LCMV Cl-13

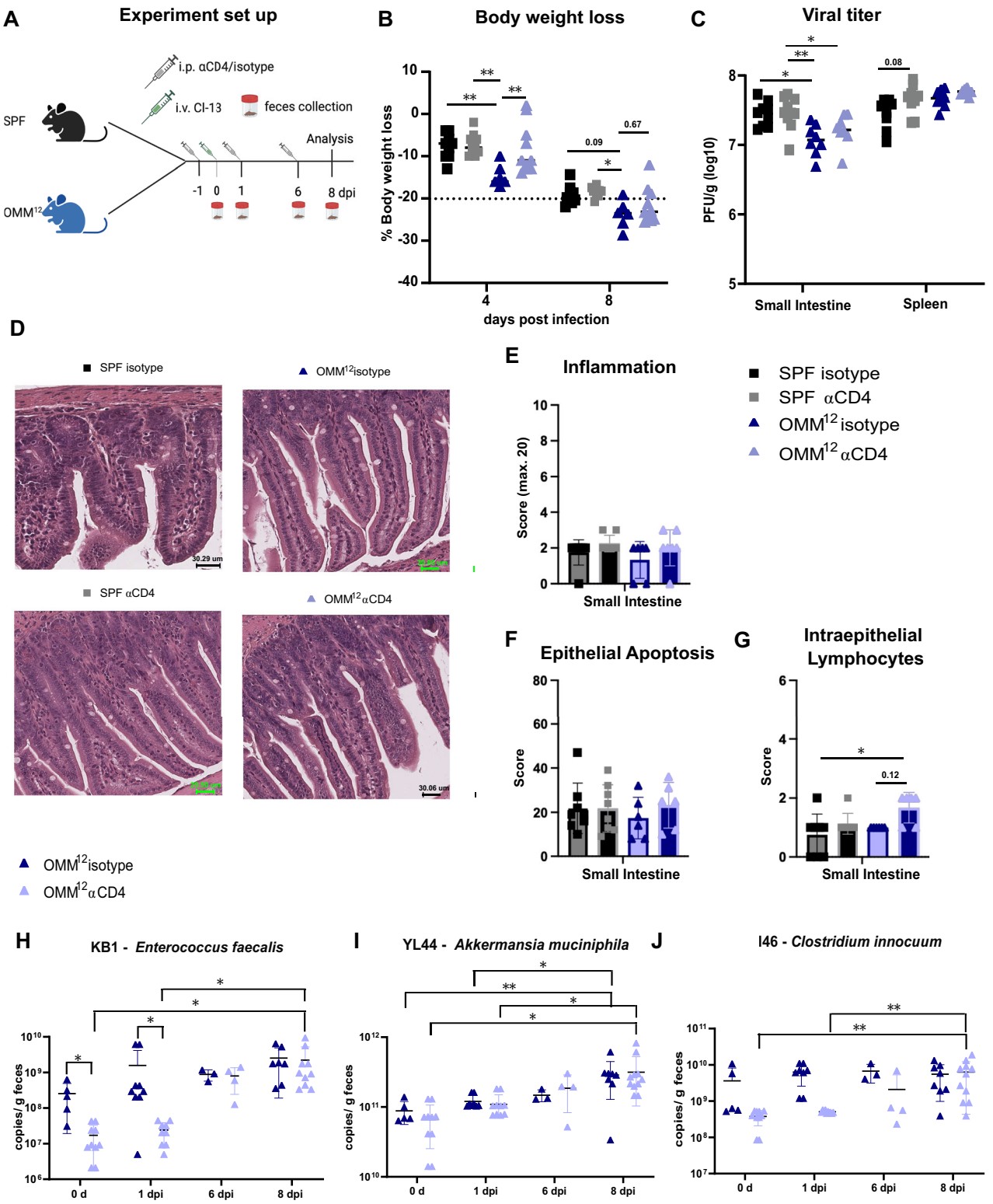

infection and re-isolated expanded SMARTA T cells from the spleen and the lamina propria of the small intestine at 8 dpi and performed single cell RNA sequencing (scRNAseq). Clustering of all isolated SMARTA T cells in a uniform manifold approximation and projection (UMAP) plot resulted in the identification of nine individual clusters (Fig. 4A). Unsupervised KNN (K-Nearest-Neighbor-Algorithm) clustering as well as cluster distribution per sample allowed the identification of differences among microbiome groups and between organs (Fig. 4B, C). Cluster 1 and 2 were almost exclusively present in SMARTA

T cells re-isolated from spleens with the highest relative abundance in SPF animals and much lower frequencies in GF and OMM[12] animals (Fig. 4B, C). Both clusters showed a substantially different expression profile compared to all other clusters with high expression of genes associated to follicular helper T (Tfh) cells (*Cxcr5, Bcl6, Il21, Id3, Il7r, Slamf6* and *Izumo1r*) (Fig. 4D, E). Moreover, both clusters lacked the expression of genes associated to Th1 effector function (*Runx3, Cxcr6, Prdm1, Ly6c2, GzmB, Gzmk, Nkg7, Ccl5* and *Ifng*) observed in clusters 3-9 (Figs. 4D, E and Supplementary Fig. 4A). None of the clusters revealed a

**Fig. 2 | CD4+ T cell depletion prevents body weight loss and viral clearance in OligOMM12 colonized animals.** SPF (gray) and OligOMM12-colonized (OMM12, blue) mice were treated one day before, on the day of the clone-13 LCMV infection and 6 days post infection (dpi) with either a depleting anti-CD4 or an isotype antibody control as illustrated in (**A**) Illustration of the experimental groups and setup. Created in BioRender (https://BioRender.com/x96c559). **B** Percentage of body weight loss of indicated groups at 4 and 8 dpi. Dotted line indicates 20% body weight loss (endpoint criteria). *P* values are as followed: for 4 dpi (SPF isotype vs OMM12 isotype = 0.0014, SPF anti-CD4 vs OMM12 isotype = 0.0061 and OMM12 isotype vs OMM12 anti-CD4 = 0.0034) and for 8 days post infection (SPF anti-CD4 vs OMM12 isotype = 0.0191). **C** Plots depict plaque forming units (PFU/g) in the small intestine and spleen of SPF and OMM12 colonized animals treated with an isotype antibody control or anti-CD4 antibody at 8 dpi. *P* values are as followed: for small intestine (SPF isotype vs SPF anti-CD4 = 0.0160, SPF anti-CD4 vs OMM12 isotype = 0.0042 and SPF anti-CD4 vs OMM12 anti-CD4 = 0.0438). **D** Representative pictures of haematoxylin and eosin-stained sections of the small intestine for the indicated groups. Bar plots show (**E**) inflammation score, (**F**) epithelial apoptosis counts **G**) and infiltration of intraepithelial lymphocytes. *P* values are for small intestine (SPF isotype vs OMM12 anti-CD4 = 0.0115). **H, I, J**) Colonization dynamics of OMM12 mice during infection and after CD4 depletion: absolute abundance of individual OMM12

strains. *P* values are as followed: for (**H**) d0 (OMM12 isotype vs OMM12 anti-CD4 = 0.00569) and for 1 dpi (OMM12 isotype vs OMM12 anti-CD4 = 0.0041) and OMM12 anti-CD4 (1 dpi vs 8 dpi = 0.0363 and d0 vs 8 dpi = 0.03296, **I**) OMM12 isotype (d0 vs 6 dpi = 0.0085 and 1 dpi vs 6 dpi = 0.0123) and OMM12 anti-CD4 (d0 vs 8 dpi = 0.016 and 1 dpi vs 8 dpi = 0.0280) and **J**) OMM12 isotype vs OMM12 anti-CD4 (1 dpi = 0.0001 and d0 = 0.0350) and OMM12 anti-CD4 (d0 vs 8 dpi = 0.0026 and 1 dpi vs 8 dpi = 0.0033). Data are plotted as 16S rRNA gene copy number of the individual strains per g of extracted gDNA. Data from two independent experiments are shown. Data from SPF (ctr) *n* = 10, SPF (anti-CD4) *n* = 10, OMM12 (ctr) *n* = 8, OMM12 (anti-CD4) *n* = 8 mice are shown (**A**–**C**). Histological analysis is shown from SPF (ctr) *n* = 8, SPF (anti-CD4) *n* = 8, OMM12 (ctr) *n* = 6, OMM12 (anti-CD4) *n* = 6 mice (**D**–**G**). Abundance of 16S rRNA copy number from OMM12 (ctrl) *n* = max. 8 and OMM12 (anti-CD4) *n* = max. 8 according to capacity of animals to donate feces (**H**–**J**). Each dot represents an individual mouse and mean ± SD from two independent experiments is shown. Statistical analysis for **B**), **C**), **F**) and **G**) was performed by using one-way ANOVA with Tukey correction for multiple comparison. Statistical analysis for (**H**–**J**) was done by performing multiple unpaired *t*-tests between each group. *P* value of < 0.05 was considered statistically significant. **p* < 0.05, ***p* < 0.01. Source data are provided as a Source Data file.

specific pattern of exhaustion markers at 8 dpi (Supplementary Fig. 4B). Cluster 3 was most prevalent in the spleen but only half as abundant in SPF compared to OMM12 and GF mice (Fig. 4C) and showed enriched gene expression of *Ly6c2, Itgb7, Ifngr1* and *S1pr4* genes (Fig. 4D). Moreover, cluster 3 highly expressed *Cxcr3* and *Tbx21* both of which genes that are associated with effector Th1 cells (Figs. 4E and Supplementary Fig. 4C, D). Additionally, we found high expression of *Cx3cr1, S1pr1, Klf2* and *Ccl5* (Fig. 4E), a profile that has already been defined as a Th1 effector cells in LCMV infection[38]. Cluster 4 was enriched for SMARTA cells of the lamina propria of SPF mice compared to other groups (Fig. 4C). This cluster was defined by high expression of *Ccl4, Ccl3, Ifng* and *Havcr2* (Fig. 4D and Supplementary Fig. 4A, B, E) which are typically expressed by effector memory and terminally differentiated T cells[39]. Given the timepoint of analysis (8dpi) we excluded effector memory cells and define cluster 4 as terminally differentiated Th1 effector cells. Cluster 5 was almost uniquely present in the small intestine with the highest prevalence in GF and OMM12 mice (Fig. 4C). This cluster was enriched in genes like *Rgs1, Plac8* and *Il21r* and showed a unique gene expression profile with increased expression of *Itga1* and *Cd7* and decreased expression of *Klf2* and *S1pr1* defining tissue-resident cells (Fig. 4D)[40]. Moreover, cells of cluster 5 expressed genes typically found in T cells present in non-lymphoid tissue genes like *Vps37b, Id2, Cxcr6* and *Ccr9* (Fig. 4D, E)[41]. Given the fact that cells of cluster 5 are exclusively derived from the lamina propria of the small intestine (Fig. 4B), cluster 5 may represent tissue-resident Th1 effector cells that are preferentially expanded in OMM12 but also GF animals. Cluster 6-9 were composed of cell subsets of less than 10% relative abundance (Fig. 4C). Cluster 6 highly expressed *Il18r*, while in cluster 7 many type I IFN stimulated genes like *Ifit1, Ifit3, Ifit3b* are expressed (Fig. 4D). A similar subset has been observed previously after LCMV infection[38]. Cells of clusters 8 and 9 showed a typical cell cycling gene expression pattern (Fig. 4D). Altogether, our scRNAseq analysis revealed a strong impact of the OMM12 consortia on the functional differentiation of virus specific CD4+ T cells post LCMV infection, namely OMM12 colonized animals favored differentiation of Th1 effector cell associated clusters at the expense of Tfh cells.

**OMM12 colonized animals show an accumulation of virus-specific cytotoxic T helper cells at the expense of follicular helper T cells**
As mentioned before, cells of cluster 1 were found to be enriched in genes reminiscent of Tfh cells[38,42,43] such as *Cxcr5, Sostdc1* and *Tox* (Fig. 4D and Fig. 5A). Cluster 1 was also enriched in transcripts of Tfh markers like *Tcf 7* and *Slamf6*, the transcription factor *Bcl6*, the

cytokine *Il21* and the inhibitory receptor *Pdcd1* (Figs.4D, E and Supplementary Fig. 4B), all of which suggesting a clear Tfh cell identity[44,45]. Cluster 2 was highly abundant in the spleen of SPF animals following the abundance trend of cluster 1 (Fig. 4B, C). Cells from this cluster showed an enrichment of multiple genes like *Slamf6, Tcf7, Id3, Itgb1* as well as the retention receptors *Ccr7* and *S1pr1* (Fig. 4D, E). Moreover, cluster 2 showed high expression of *Klf2* and *Il7r* which typically restrain Tfh cell generation[46,47] (Fig. 4E) and relative to cluster 1 a lower expression of the Tfh-associated genes *Bcl6* and *Cxcr5* (Fig. 5A) and may thus represent Tfh progenitor cells. Cluster 1 and 2 were almost uniquely derived from spleen and greatly diminished in OMM12 and GF compared to SPF animals (Fig. 4D and Fig. 5A–C). To confirm the relative reduction of Tfh cell differentiation in gnotobiotic animals on protein level, we additionally analyzed SMARTA T cells at 8 dpi for Tfh markers. Indeed, SMARTA cells from gnotobiotic and especially OMM12 animals showed a reduced frequency in cells with high CXCR5 expression (Fig. 5B, C). Thus, virus-specific T cells in OMM12 animals may favor an alternative differentiation path compared to SPF animals.

One of the hallmarks of LCMV Cl-13 infection in OMM12-colonized animals was the elevated expansion and accumulation of virus-specific T cells in the small intestine (Fig. 3). Cluster 5 probably representing tissue-resident Th1 effector cells was the only cluster that might explain the enhanced accumulation of LCMV-specific T helper cells. Interestingly, cluster 5 showed high expression of the *Plac8* gene that was highest in the small intestine of OMM12 animals (Fig. 5D). *Plac8* is uniquely expressed by Th1 effector cells compared to all other CD4+ T cell subsets and is crucial to regulate Th1 and CD8+ T cell response during Th1 driven inflammation[48]. Furthermore, *Plac8* overexpression has been associated with cell growth and migration[49], possibly explaining the higher proliferation of SMARTA T cells in the lamina propria of the small intestine of OMM12 animals (Fig. 3E). Cells of cluster 5 also expressed high levels of the chemokine receptor *Ccr9* required for migration of lymphocytes to the intestinal tract (Fig. 5E)[50,51]. Besides homing and general features of cell proliferation cells of this cluster also expressed lower levels *Ly6c2* and intermediate levels of *Tbx21* (Fig. 5F), a profile associated to an enhanced ability for proliferation and longevity of Th1 effector cells[52]. Altogether, we determined to refer to cluster 5 as proliferative tissue-resident effector Th1 cells. In contrast, cytotoxic Ly6c2hi effector Th1 cells[52] could be linked to cluster 3 with uniquely high expression of *Ly6c2* (Fig. 5F) indicative of an increased capacity for GzmB and IFNγ-associated cytotoxicity. However, *Ifng* was expressed at higher levels in SMARTA T cells re-isolated from the small intestine compared to the spleen (Fig. 3F, I). This was most pronounced in cells isolated from OMM12

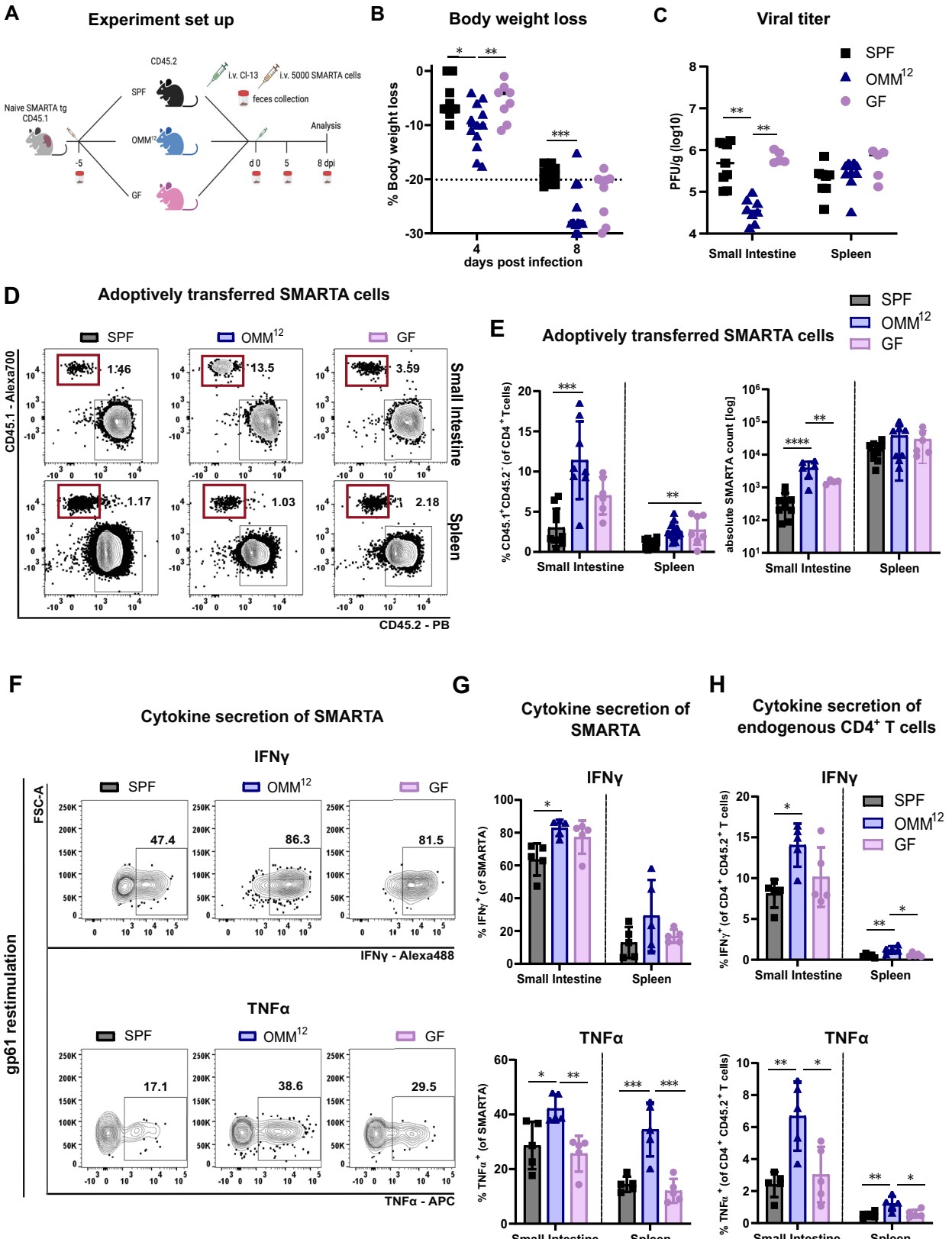

animals compared to SPF confirming our observations by flow cytometry (Figs. 3F–G and Supplementary Fig. 4E). Furthermore, *GzmB* expression was increased in cluster 3 in the spleen again particularly pronounced in the OMM[12] animals (Fig. 5G). This was confirmed on the protein level by re-stimulating adoptively transferred SMARTA T cells of OMM[12] animals with the virus peptide gp61 (Fig. 5H, I). In summary,

these results indicate that the intestinal OMM[12] microbiome cannot only influence locally the accumulation of tissue-resident T helper cells but possibly also effector functions of virus-specific T helper cells such as cytotoxicity at other organs. Thus, a stable and defined minimal bacterial consortium is able to shape the immune response to a systemic virus infection via enhancing effector cell function of CD4+ T

**Fig. 3 | Virus-specific T helper cells (SMARTA) show an increased expansion and effector function in OligOMM[12]-colonized mice. A** $5 \times 10^3$ SMARTA T cells were adoptively transferred into SPF (gray), OligOMM[12]-colonized (OMM[12], blue) and germ-free (GF, pink) mice 5 days prior infection with LCMV clone-13 and analyzed at 8 days post infection (dpi). Created in BioRender ([https://BioRender.com/w55l495](https://BioRender.com/w55l495)). **B** Percentage of body weight loss of indicated groups at 4 and 8 dpi. Dotted line indicates 20% body weight loss (endpoint criteria). *P* values are as followed: 4 days post infection (SPF vs OMM[12] = 0.0491 and OMM[12] vs GF = 0.00462), 8 days post infection (SPF vs OMM[12] = 0.0003). **C** Plots depict plaque forming units (PFU/g) in the small intestine and spleen of GF, SPF and OMM[12] colonized animals at 8 dpi. *P* values are as followed: small intestine (SPF vs OMM[12] = 0.0020 and OMM[12] vs GF = 0.0057). **D** Representative flow cytometry plots and (**E**) frequencies as well as absolute cell numbers of CD45.1[+] SMARTA T cells in indicated organs. *P* values are as followed: for frequencies of SMARTA in small intestine (SPF vs OMM[12] = 0.0001) and spleen (SPF vs. GF = 0.0053); for the absolute SMARTA count in small intestine (SPF vs OMM[12] < 0.0001 and OMM[12] vs GF = 0.0027). Representative flow cytometry plots (**F**) and (**G**) frequencies of cytokine[+] SMARTA T cells after restimulation with the virus-peptide gp61 in the indicated organs. *P* values are as followed: IFNγ small intestine (SPF vs OMM[12] = 0.0108), TNFα small intestine (SPF vs OMM[12] = 0.0236, OMM[12] vs GF = 0.0069) and spleen (SPF vs OMM[12] = 0.0009 and OMM[12] vs GF = 0.0003). **H** Frequencies of cytokine[+] endogenous CD4[+] T cells after restimulation with the virus-peptide gp61 in the indicated organs. *P* values are as followed for IFNγ small intestine (SPF vs OMM[12] = 0.0143) and spleen (SPF vs OMM[12] = 0.0081 and OMM[12] vs GF = 0.0196) and for TNFα small intestine (SPF vs OMM[12] = 0.0042 and OMM[12] vs GF = 0.0119) and spleen (SPF vs OMM[12] = 0.0081 and OMM[12] vs GF = 0.0196). For small intestine samples as well as cytokine stimulation samples of two animals of the same group were occasionally pooled to increase the yield. Each dot represents an individual mouse or two if pooled and mean ± SD from two independent experiments is shown. Data from two independent experiments with (**B**) SPF *n* = 12, OMM[12] *n* = 12, GF *n* = 10 mice, (**C**) SPF *n* = 9, OMM[12] *n* = 8, GF *n* = 5 or (**E**−**G**) SPF *n* = 5-10, OMM[12] *n* = 5-10, GF *n* = 5-7 mice are shown. Statistical analysis for (**B**, **C**, **E**, **G** and **H**) was performed using one-way ANOVA with Tukey correction for multiple comparison. *P* value of < 0.05 was considered statistically significant. *$p < 0.05$, **$p < 0.01$, ***$p < 0.001$, ****$p < 0.0001$. Source data are provided as a Source Data file.

---

helper cells including cytotoxic T cell subsets and viral clearance at the cost of enhanced disease pathology.

## Discussion

Here, we established the systemic LCMV Cl-13 infection in gnotobiotic mice and investigated adaptive anti-viral immunity by tracking adoptively transferred virus-specific CD4[+] T cells. To our knowledge, only few reports investigated the role of the microbiome for the immune response to a systemic infection in such a highly controlled manner. We observed an enhanced body weight loss manifesting early after infection and improved viral clearance in the intestine of OMM[12]-colonized mice compared to SPF and GF mice. Noteworthy, this effect could be rescued very early after infection (4 dpi) by depletion of CD4[+] T cells suggesting that this OMM[12] microbiome-intrinsic effect is present early or even prior to LCMV infection. Yet, uninfected OMM[12]-colonized animals do not differ from SPF animals in body weight (Supplementary Fig. 1A and[24,25]) indicating that enhanced body weight loss in OMM[12] is triggered by the immune response after LCMV Cl-13 infection. One possibility could be that microbial signals of the microbiome influence the innate immune compartment, which in turn alter virus-specific CD4[+] T cell differentiation. The function of many different innate immune cells has been associated with the microbiome and may have the capacity to influence T helper cell differentiation. For instance, intestinal dendritic cells have been shown to be poised by microbiota-dependent induction of type 1 interferons, that may result in a different capacity to activate T helper cells post viral infection[53]. A previous study has already shown that mice, which were prior to and throughout the infection treated with broad-spectrum antibiotics exhibited impaired responsiveness to type I and type II IFNs show a reduced capacity to control LCMV replication[54].

NK cell responses are another cytolytic cell type that are influenced by commensal bacteria and contribute to the antiviral defense. Previous studies have shown that the priming or licensing and cytolytic activity of NK cells is compromised in GF mice[55]. Also in this study, mononuclear phagocytes derived from GF or antibiotic-treated animals were deficient in the production of type 1 interferons, which is a necessary signal for the priming of NK cells[55]. Thus, intestinal bacteria may provide tonic signals that calibrate the activation threshold and sensitivity of innate immunity. However, most of these studies solely used either GF mice to assess the general importance of the intestinal microbiome or broad spectrum antibiotic treatment regimens that may result in divergent results presumably reflecting the high variability of intestinal microbiomes across different animal facilities. Importantly, we describe here an effect of a defined minimal microbial consortium that is not apparent in GF animals and thus excludes a pure lack of such tonic microbial signals as causative underlying effect.

Moreover, and in contrast to previous reports, we were able to show a clear microbiome effect on virus-specific T helper cells: OMM[12]-colonized animals showed a superior expansion of adoptively transferred LCMV-specific CD4[+] T helper cells and an increased effector function. This effect was particularly pronounced in the lamina propria of the small intestine, emphasizing a possible local effect of the intestinal microbiota on resident or circulating T helper cells. Flow cytometry and single cell RNAseq analysis of adoptively transferred virus-specific CD4[+] T cells revealed a relative expansion of clusters with high proliferative, cytotoxic and effector potential (clusters 3 and 5). At the same time, we observed a relative paucity of the differentiation of naïve virus-specific T cells into Tfh cells (cluster 1 and 2) in OMM[12] animals. This paucity might be explained by the deviation of T cell differentiation into Th1-associated effector cells by the influence of the OMM[12] consortium or due to an inhibitory effect of the OMM[12] microbiota on Tfh cell differentiation itself. Characterization of T helper cells in gnotobiotic mouse models at steady state is often restricted to Th17 or Treg induction induced by the respective consortium but fails to quantify potential effects on Th1 cell differentiation[25,30]. However, OMM[12] animals succumb rapidly to severe disease pathology possibly due to the massive expansion of Th1 effector cells with a cytotoxic profile and Tfh cells have shown to be essential only in later stages of LCMV Cl-13 infection[56,57]. Importantly, vaccine-elicited anti-viral CD4 T cells have been previously shown to be principally capable of causing a fatal disease pathology after chronic LCMV infection[58].

Due to the severe body weight loss that we observed in OMM[12] mice, we were not able to conduct any experiments with LCMV Cl-13 beyond the acute phase (day 8 after infection). The use of other chronic LCMV strains like Docile with less severe disease pathology may be helpful to elucidate the impact of the microbiome on Tfh differentiation and antibody responses with higher relevance at later time-points after infection[13]. It remains possible that the OMM[12] consortium contains a higher frequency of antigens that have features of molecular mimicry for LCMV-specific epitopes and may thereby exert some priming effect already prior to an LCMV infection.

Another surprising finding from our study was that CD8[+] T cells were almost unaffected by the microbiome status of the animals while we identified cytotoxic and inflammatory CD4[+] T cell subsets as the main player for disease pathology in OMM[12] animals[29]. Previous studies revealed that cytotoxic and inflammatory CD4[+] T helper cells can be crucial for viral clearance of LCMV[59,60], Ectromelia virus[61], Influenza A virus[62], West Nile virus[63], murine Gammaherpesvirus 68[64] but can also induce fatal immunopathology in a vaccination model and a

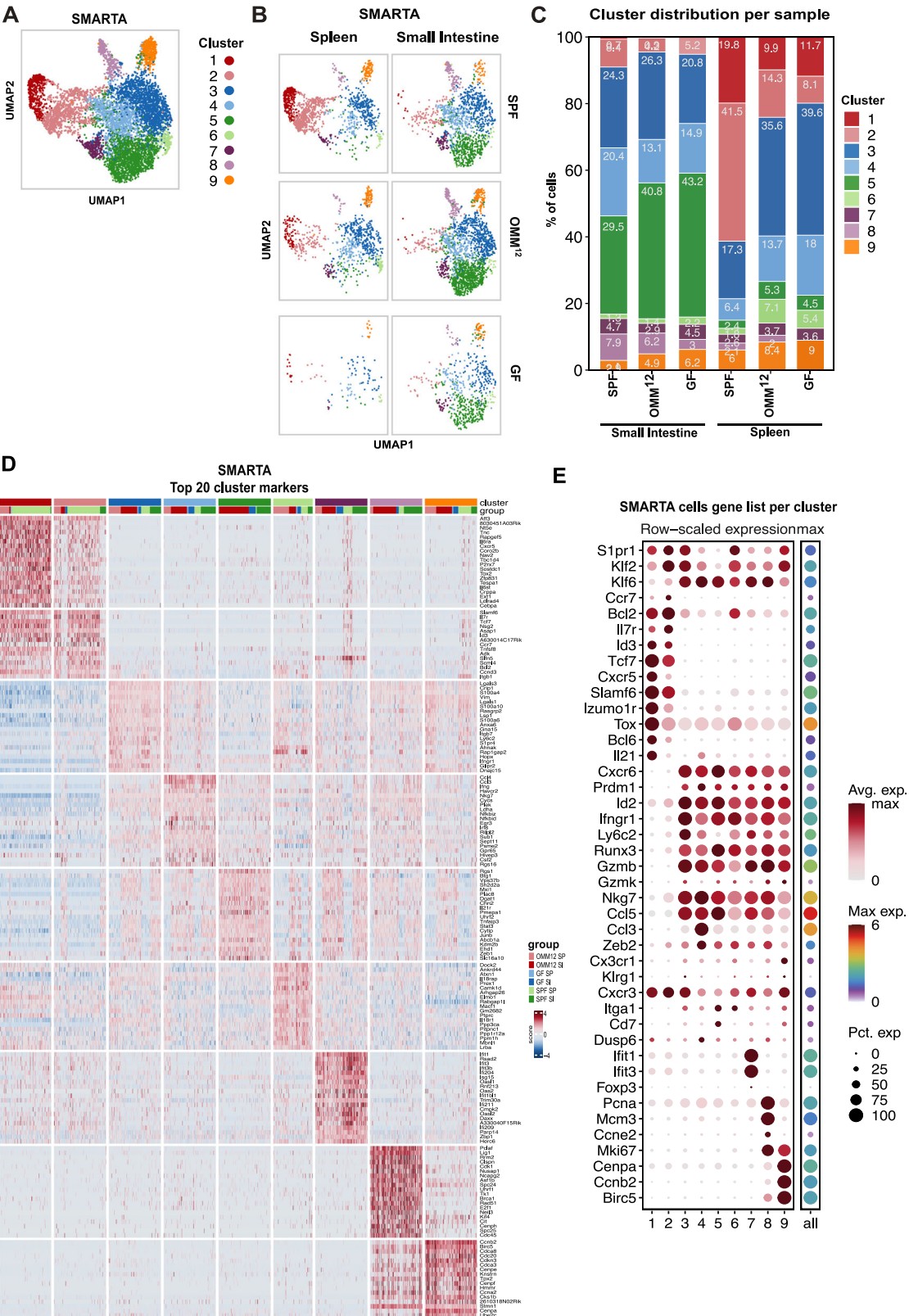

**Fig. 4 | SMARTA cells are transcriptionally different among SPF, OMM12 and GF mice.** SPF (gray), OligOMM12-colonized (OMM12, blue) and germ-free (GF, pink) mice received 5000 adoptively transferred SMARTA T cells 5 days prior infection with LCMV Cl-13. SMARTA T cells (live/dead− CD4+CD45.1+) cells from small intestine lamina propria and spleen of 3–5 mice per group were re-isolated at 8 dpi and subjected to RNAseq analysis. **A** UMAP depicting Louvain cluster for all cells or (**B**) for each organ and microbiome condition individually. **C** Relative distribution of clusters in each sample group. Numbers inside the bars indicate the percentage of cells within a specific cluster. **D** Heatmap of expression levels for the topmost significant cluster-defining transcripts. **E** Feature plot of the average expression and percentage of cells from each cluster for selected genes.

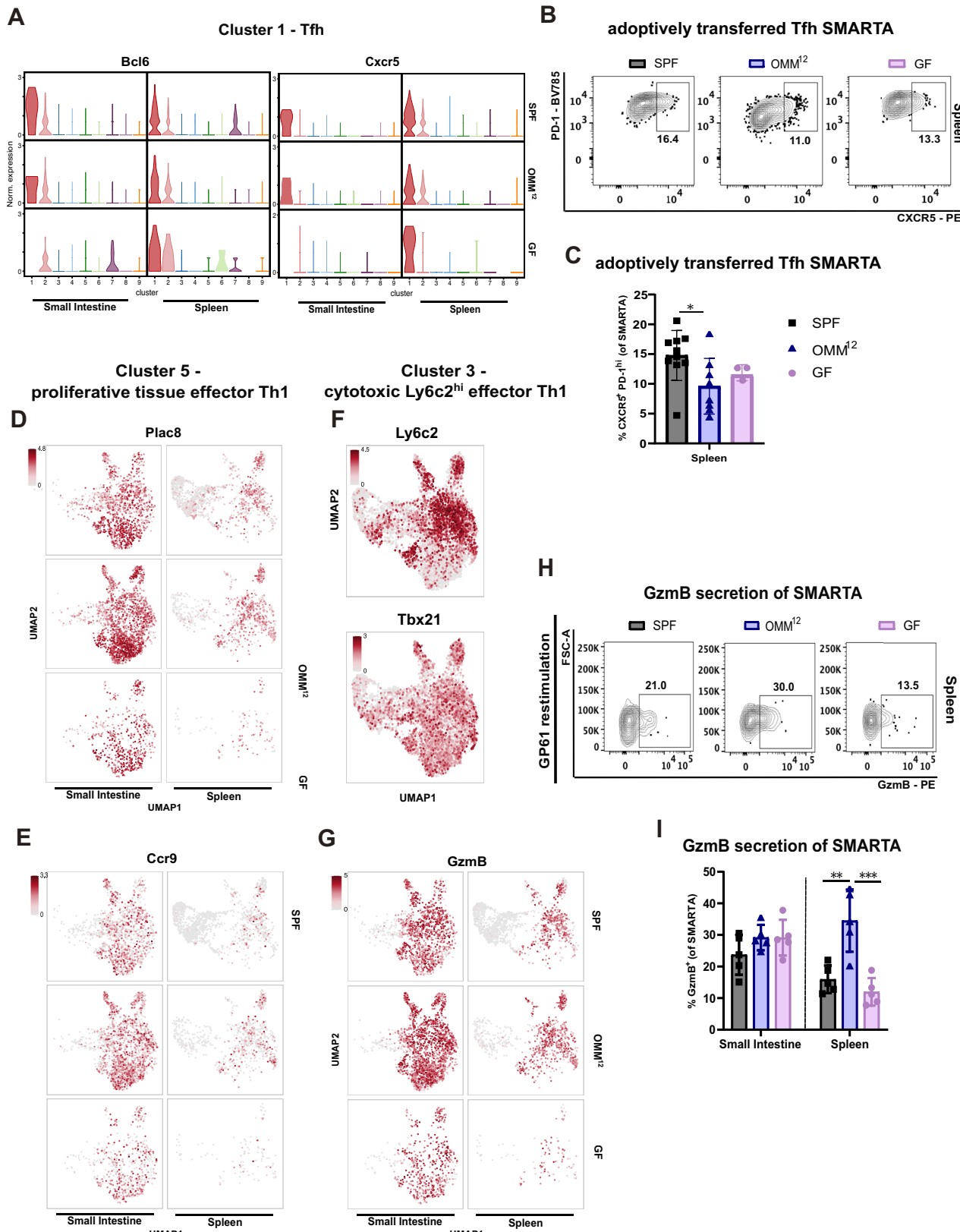

subsequent chronic LCMV infection[58]. It is important to note that the OMM[12] microbiota remained stable during the course of LCMV infection. This observation suggests that a systemic pro-inflammatory response does not allow for the outgrowth of single bacteria from OMM[12] with a dominant effect on anti-viral immunity. Furthermore, we also ruled out enhanced inflammation in the lamina propria of OMM[12]

animals as a general cause for stronger virus specific T helper cell responses because histological examination found no evidence for intestinal inflammation in OMM[12] mice. Rather, bacterial products present in the OMM[12] consortium may act either directly on virus-specific T helper cell proliferation and differentiation or indirectly prime innate immune cells which then regulate T helper cell fate. For

**Fig. 5 | OMM$^{12}$ favors cytotoxic and proliferative Th1 cell clusters at the expense of Tfh cells.** SPF (gray, square), OMM$^{12}$ (blue, triangle) and GF (pink, circle) mice received 5.000 SMARTA cells 5 days prior infection with LCMV Cl-13 and were analyzed at 8 dpi. Small intestine lamina propria and spleen live$^+$ CD4$^+$ CD45.1$^+$ cells were re-isolated at 8 dpi and subjected to RNAseq analysis. Expression of (**A**) *Bcl6* and *Cxcr5* on each individual cell for each sample group and organ represented over the reduced space of Violin plots. Representative (**B**) FACS plots and (**C**) frequencies of adoptively transferred Tfh SMARTA cells in spleen of SPF, OMM$^{12}$ and GF mice. *P* values are as followed: for adoptively transferred Tfh SMARTA in the spleen (SPF vs OMM$^{12}$ = 0.0043). Expression of (**D**) *Plac8*, (**E**) *Ccr9*, (**F**) *Ly6c2* and *Tbx21* and (**G**) *GzmB* on each individual cell for each sample group represented over

the reduced space of UMAP. **H** Representative flow cytometry plots and **I**) frequencies of GzmB secretion of adoptively transferred SMARTA cells after restimulation with the virus-specific gp61 peptide. *P* values are for GzmB secretion of SMARTA in spleen (SPF vs OMM$^{12}$ = 0.0015 and OMM$^{12}$ vs GF = 0.0005). Each dot represents an individual mouse and mean ± SD from two independent experiments is shown. Data from two independent experiments with SPF *n* = 10, OMM$^{12}$ *n* = 8, GF *n* = 3 mice (**C**) or SPF *n* = 5, OMM$^{12}$ *n* = 5, GF *n* = 5 mice (**I**) are shown. Statistical analysis for (**B**, **C**, **E**, **G** and **H**) was performed using one-way ANOVA with Tukey correction for multiple comparison. *P* value of < 0.05 was considered statistically significant. \**p* < 0.05, \*\**p* < 0.01, \*\*\**p* < 0.001. Source data are provided as a Source Data file.

instance, butyrate can directly affect the accessibility and expression of genes associated with Th1 cell differentiation like IFNγ and T-bet[65,66]. This metabolite is produced by three members of the OMM$^{12}$ community: *F. plautii* YL31, *C. innoccum* YL32 and *E. clostridiforme* YL31. Two of them, *F. plautii* YL32 and *C. innoccum* I46, showed a trend of increased abundance during LCMV Cl-13 infection (Figs. 2G and Supplementary Fig. 2M). Increased abundance of *C. innocuum* has been previously associated with intestinal and extra-intestinal infections[67]. Interestingly, *A. muciniphila* has been linked to efficacy of checkpoint therapy in anti-cancer treatments and has been associated with enhanced T helper cell recruitment and activation and/or Th1 effector cell differentiation[68,69]. Recently, the effect of OMM$^{12}$ colonization across sex and age on the metabolome in various tissues was defined and this resource may help to identify many more OMM$^{12}$ metabolites or indirect effects involved in shaping the immune response to LCMV infection[70]. Defined minimal bacterial consortia are a helpful tool to identify such factors or metabolites[71]. For example, comparing immune responses to LCMV in OMM$^{12}$ derivates with either additional bacteria or selective elimination of individual bacterial strains from OMM$^{12}$ [30]. The identification of molecules and bacterial compounds is particularly relevant for the translation of our findings to human diseases and, hopefully, to improve the responsiveness of vaccination and treatment regimens to chronic viral infections. Recent insights on the important role of the intestinal microbiome for the responsiveness towards checkpoint inhibition as a treatment of certain cancer types or to predict graft-versus-host disease[68,69,72,73]. Many reports clearly demonstrated an essential role of the intestinal microbiome in influencing Th17 and particularly regulatory T cells[74]. In the tumor microenvironment, the latter are often inversely correlated to Th1 and cytotoxic T cells[75,76]. It is tempting to investigate the function of the microbiome or the metabolites they produce also directly on the differentiation and effector function of Th1 cells present in tumors[77]. Immune responses to viral infections are not directly comparable to the complex microenvironment found in tumors but may serve as an ideal screening platform to identify such interactions. The limited complexity and the highly controlled conditions of gnotobiotic mouse models will hopefully allow in the future the identification of single molecules exerting such profound effects on T helper cells. In this work, we have shown a clear effect of the microbiome on the adaptive anti-viral immune response which has critically influenced viral infection induced disease pathology. Nevertheless, there are some inherent limitations of our study that limit translation of our findings to humans such as the use of an animal model in itself, the use of a rodent-borne virus and a minimal bacterial flora based on murine commensal species. Moreover, we acknowledge the need to study the impact of the OMM$^{12}$ bacterial consortium on innate immune cells as putative accessory cells to translate microbiome-derived modulation of T helper cell responses. Identifying the mechanistic link of the microbial connection with immune cells will further reveal the

therapeutic potential. Moreover, this could enable renouncing fecal microbial transplantation, live biotherapeutic products or probiotics that can become problematic and instead move into better targeted therapeutics like the administration of metabolites to improve anti-viral immunity.

## Methods

### Animals

C57BL/6 SPF mice (CD45.2$^+$) animals were purchased from Charles River (CRL France) and maintained under specific pathogen-free (SPF) conditions at the Technical University of Munich, Germany. SMARTA TCRαβ transgenic mice backcrossed to CD45.1 background were kindly provided by A. Oxenius[78] and bred and maintained under SPF conditions at the Technical University of Munich, Germany. Germ free (GF) mice and gnotobiotic mice stably colonized with the Oligo-Mouse-Microbiota 12 (OMM$^{12}$) mice were bred and maintained at the Central Animal Facility of Hannover Medical School, Germany) according to standard operating procedures[79]. OMM$^{12}$ mice harbor the following strains: *Acutalibacter muris* KB18, *Flavonifractor plautii* YL31, *Enterocloster clostridioforme* (former *Clostridium clostridioforme*) YL32, *Blautia coccoides* YL58, *Clostridium innocuum* I46, *Limosilactobacillus reuteri* (former *Lactobacillus reuteri*) I49, *Enterococcus faecalis* KB1, *Bacteroides caecimuris* I48, *Muribaculum intestinale* YL27, *Bifidobacterium animalis* YL2, *Turicimonas muris* YL45 and *Akkermansia muciniphila* YL44[23]. Gnotobiotic mice were kept in sterile isolators throughout the course of the experiments and received pelleted 50 kGy gamma-irradiated feed (Ssniff Spezialdiäten, Soest, Germany) and autoclaved water *ad libitum*. Mice housed in isolators were constantly monitored according to recommendations for maintaining gnotobiotic colonies[80] and FELASA recommendations[81] and were routinely tested for contaminations. Samples were collected under sterile conditions using sterile forceps and placed into sterile tubes to minimize the risk of contamination by environmental bacteria and fungi during sampling. All interventions were carried out with age- and sex-matched at least 6-week-old male and female mice (according to availability) in compliance with the Technical University of Munich institutional regulations and were approved by local ethics committee and the appropriate government authority (Regierung von Oberbayern, license number ROB-55.2-2532.Vet_02-19-137). Endpoint was defined by a score composed of different weighted criteria including body weight, body conditions, behavior and general condition. All mice were exposed to a 12:12 h light-dark cycles with food and water administration *ad libitum*. Animals were randomly assigned to experimental groups which were non-blinded, and no specific method was used to calculate sample sizes.

### Histological scoring

For histological examinations, the small intestines of the animals were collected, fixed in paraformaldehyde (4%) and subjected to embedding in paraffin using standard protocols. Hematoxylin-Eosin-stained slides were produced from the paraffin blocks and evaluated by an experienced board-certified pathologist (KS). Inflammation scoring was performed using an established scoring system for small intestinal

inflammation[82]. This scoring system takes the inflammatory cell infiltrate (severity and extend) and epithelial changes as well as mucosal architecture into account. The maximal score is 20. Additionally, the average number of intraepithelial lymphocytes migrating through the villous epithelium were counted within 5 high-power fields and the total number of epithelial apoptosis within 5 high-power fields were counted. These analyzes were performed on an Olympus microscope (BX53, field number 22). All slides were scanned with an Aperio AT2 scanner and representative images were taken using ImageScope v. 12.4.6.5003.

## 16S RNA gene sequencing
Fecal pellets were weighed and frozen at -80 °C. On the day of processing, samples were thawed, and the DNA was extracted with the Maxwell RSC fecal microbiome DNA kit (Promega, #AS1700) according to the manufacturer's protocol. The 16S rRNA gene was amplified by PCR using barcoded primers flanking the V3 and V4 hypervariable regions (CCT ACG GGN GGC WGC AG and GAC TAC HVG GGT ATC TAA TCC)[83,84]. Sequencing was performed on an Illumina MiSeq platform according to the manufacturer's instructions. The generated FASTQ files were demultiplexed and transformed into zero-radius Operation Taxonomic Unit (zOTU) tables using the Integrated Microbial Next Generation Sequencing platform (trimming of ten nucleotides at the 5' and 3' end, abundance cut-off of 0.25%). Further analysis was done by using the R pipeline Rhea[85] and Namco[86]. Alpha-Diversity was calculated as species richness and Shannon effective number of species. Distance matrix for beta-diversity was calculated based on the generalized UniFrac approach[87].

## DNA extraction from feces
gDNA extraction was performed using a phenol-chloroform protocol. Briefly, fecal pellets were resuspended in 500 µl extraction buffer (200 mM Tris-HCl (Sigma-Aldrich, #E884), 200 mM NaCl, 20 mM EDTA (Sigma-Aldrich, #E884) in ddH2O, pH 8, autoclaved), 210 µl 20% SDS and 500 µl phenol:chloroform:isoamylalcohol (25:24:1, pH 7.9). Furthermore, 500 µl of 0.1 mm-diameter zirconia/silica beads (Carl Roth. #N033.1) were added. Bacterial cells were lysed with a bead beater (TissueLyser LT, Qiagen) for 4 min, 50 Hz. After centrifugation (14,000 x g, 5 min at RT), the aqueous phase was transferred into a new tube, 500 µl phenol:chloroform:isoamylalcohol (25:24:1, pH 7.9) were added and again spun down. The resulting aqueous phase was gently mixed with 1 ml 96% ethanol and 50 µl of 3 M sodium acetate by inverting the tube. After centrifugation (30 min, $14,000 \times g$, 4 °C), the supernatant was discarded and the gDNA pellet was washed with 500 µl ice-cold 70% ethanol and again centrifuged ($14,000 \times g$, 4 °C; 15 min). The resulting gDNA pellet was resuspended in 100 µl Tris-HCL pH 8.0 (Sigma-Aldrich, #10812846001). Subsequently, gDNA was purified using the NucleoSpin gDNA clean-up kit (Macherey-Nagel, #7420230.50) and stored at −20 °C.

## Quantitative PCR for bacteria
Quantitative PCR for the OMM[12] was performed according to[23] using the following protocol: Duplex-assays were established and optimized in a Roche LighCycler96 system. DNA extracted from feces was diluted in water (Gibco) to a final concentration of 2 ng/µl. 2.5 µl of the samples were added as duplicates in 96 well plates (Roche) mixed with 0.2 µl of respective primers (see Supplementary Table 1) (30 µM) and hydrolysis probe (25 µM), 10 µl 2x FastStart Essential DNA Probes Master (Roche, #4913949001) and 5.5 µl H2O (Gibco). The following cycler conditions were used: preincubation for 10 min at 95 °C, followed by 45 cycles of 15 s 95 °C and 60 s 60 °C. Fluorescence was recorded after each cycle. Standard curves using linearized plasmids containing the 16S rRNA gene sequence of the individual strains were used for absolute quantification of 16S rRNA gene copy numbers of individual strains. Data were analyzed with the LightCycler96 software package (Roche).

## Viral infection
The LCMV clone-13 (Cl-13) strain was propagated in baby hamster kidney cells and titrated on Vero African green monkey kidney cells according to established protocols[88]. 6–8 week old mice were injected intravenously with $2 \times 10^6$ plaque forming units (PFU) of the Cl-13 strain. Chronic state of infection 8 dpi was routinely verified by PD-1 staining on CD8 T cells.

## CD4 T cell depletion
For the depletion of CD4+ lymphocytes in vivo, 300 µg anti-mouse CD4 antibody (clone GK1.5, BioXCell) or the IgG control antibody (BioXCell) was diluted in PBS and injected intraperitoneally on day −1, day 1 and day 6 post infection. Successful depletion was ensured by flow cytometry staining of splenocytes.

## Adoptive cell transfer
For adoptive T cell transfer, splenocyte cell suspensions of CD45.1+ SMARTA transgenic mice were treated with a hypotonic ammonium-chloride-potassium (ACK) buffer for red blood cell lysis. CD4+ T cells were purified with a mouse CD4+ T cell enrichment kit (Miltenyi Biotech, #130-104-454) according to the manufacturer's instructions. Cell purity was routinely checked by flow cytometry by staining (CD45.1+CD4+TCRVa2+CD44−). Purified SMARTA T cells were washed extensively with sterile PBS (Sigma-Aldrich, #D8537) and $3 - 5 \times 10^3$ CD45.1+ SMARTA T cells were injected intravenously one to five days prior to infection with LCMV clone 13.

## Viral titers
Viral load was determined by standard plaque assay on Vero cells based on the protocol of Battegay[88]. Organs were harvested, weighed and immediately frozen in serum free media (RPMI 1640, Sigma-Aldrich, #R7755) at -80 °C until further processing. Frozen tissue was homogenized in serum free media (RPMI 1640, Sigma-Aldrich, #R7755 and centrifuged 15 mins $15000 \times g$ at 4 °C. Five-fold dilutions from the supernatant were added into a 24 well plate kept on ice and $2.5 \times 10^5$ Vero cells/well were added on top. Plates were then incubated at 37 °C in a CO2 incubator for four hours, allowing the cells to settle and adhere to the plates. Lastly, 400 µl of a 1:1 mixture of 2% methylcellulose (Sigma-Aldrich, #M0262) in 2 x DMEM (Sigma-Aldrich, #D5648) with 10% FCS and glutamine was added. After the infection, plates were incubated for 40–48 h at 37 °C in a CO2 incubator. Then, the media was aspirated and washed with PBS (Sigma-Aldrich, #D8537). The cell layer was fixed with 4% formaldehyde for 30 min, followed by incubation with a 0.5% Triton X-100 (Fluka, #93418) solution in PBS for another 20 min to permeabilize fixed cells. Nonspecific binding was blocked by an incubation step of one hour with PBS containing 10% FCS. The samples were then overlayed with 200 µl of anti-LCMV antibody (clone VL-4, BioXcell) and incubated for 60 min, followed by a 60 min incubation with 200 µl peroxidase-labeled goat anti-rat antibody (Jackson ImmunoResearch, #112-035-003). Between incubations steps plates were washed twice with PBS. Lastly, plates were incubated at room temperature for 20 min with 300 µl of a developing substrate (0.2 M Na2HPO4x2H2O, 0.1 M citric acid, 50 ml of double distilled H2O and 40 mg of s-phenylenediamine (Sigma-Aldrich, #P3888). Plaques were then manually counted, and PFU/mg was calculated.

## Cell isolation from tissues for FACS analysis
Mice were euthanized by cervical dislocation and small Intestine and spleen were harvested. Splenocytes were isolated by manually mashing spleens through a 100-µm nylon cell strainer (BD Falcon). Red blood cells were lysed with a hypotonic ACK lysis buffer (0.15 M ammonium chloride (Sigma-Aldrich, #A4514), 10 mM potassium hydrogen carbonate (Merck, #237205), 1 mM EDTA-disodium, pH-adjusted to 7.3). Cell suspensions were then washed with RPMI medium (RPMI-1640, Sigma-Aldrich, #R7755 supplemented with 10% fetal

bovine serum, 2 mM L-glutamine, 1% penicillin-streptomycin, 1 mM sodium pyruvate, 50 nM β-mercapthoethanol) and then centrifuged 5 min at 500 × $g$ at 4 °C. Splenocytes were resuspended and used for flow cytometry staining. For cell isolation of the lamina propria, the first 11 cm of the small intestine (duodenum) were used to isolate lamina propria lymphocytes[3]. Peyer's Patches were removed, and the intestine was flushed with cold PBS. The tissue was then cut longitudinally and incubated for 30 min in 30 mM EDTA in PBS at pH 8 on ice. The tissue was then shaken vigorously in a repetitive manner and washed with PBS and the supernatant was removed until the solution appeared clear. Remaining tissue was minced into small pieces and digested in RPMI containing 25 mM HEPES (Sigma-Aldrich, #C-40010), 0.05 mg/ml collagenase D (Roche, #11088866001), and 10 µg/ml DNase I (Sigma-Aldrich, #10104159001) at 37 °C for 30 min for two consecutive rounds. Between incubation steps, the tissue was pipetted several times up and down. The supernatant was transferred to a new tube and replaced with new digestion media. The combined supernatant was then filtered through a 70-µm cell strainer and centrifuged at 500 × $g$ for 10 min. The cell pellet was then resuspended in 40% Percoll (Sigma-Aldrich, #P4937) solution and layered onto an 80% Percoll layer. The Percoll gradient was run at 1500 × $g$ at room temperature for 15 min. The interlayer containing lamina propria mononuclear cells was collected and washed prior to further analysis.

### Flow cytometry

For intracellular cytokine staining, isolated lymphocytes were resuspended in complete RPMI and cultured in 96-well U-bottom plates (3 × $10^6$ cells per well) at 37 °C with 5% $CO_2$ for 30 min and stimulated at 2 µg/ml with the LCMV GP$_{61-80}$ peptide (Eurogentec, #AS-64851), 10 ng/ml PMA (Sigma-Aldrich, #P1585) and 1 µM ionomycin (Sigma-Aldrich, #I3909) or kept in complete RPMI as unstimulated control. Then, 7 µg/ml Brefeldin A (Sigma-Aldrich, #B7651) was added, and cells were incubated for another 3 h. Cells were harvested and washed with PBS. In some cases, samples from two animals of the same group were pooled to reach the appropriate number of cells.

Intracellular staining was routinely performed with the Foxp3/transcription factor staining kit (eBioscience, #00-5523-00) according to the manufacturer's instructions. Before staining, all cell preparations were incubated with 3.3 µg/ml rat anti-mouse CD16/32 (Fc receptor block) for 10 min on ice to block unspecific antibody binding. For extracellular staining, the following antibodies were used: anti-mouse CD45.1 Alexa Fluor 700 (Ly5.1, Southern Biotec), CD45.2 Pacific Blue (Ly5.2), TCRβ Super Bright 600 (H57-597), CD4 Brilliant Violet 711 (RM4-5, Biolegend), PD-1 Brilliant Violet 785 (29F-1A12). Live/dead staining was routinely done using the Zombie Aqua fixable viability kit (Biolegend, #423101). Cells were incubated with extracellular antibodies 30 min on ice and then washed twice with PBS. Cells were then fixed in 150 µl of the provided Fixation/Permeabilization buffer from the Foxp3/transcription factor staining kit overnight at 4 °C. Before proceeding with intracellular staining, cells were washed again twice with PBS. For intracellular staining the following antibodies were used: anti-mouse IFNγ-Alexa Fluor 488 (XMG1.2), Granzyme B-PE (QA16A02), TNFα-APC (MP6-XT22), T-bet-PE-Cy7 (4B10). The full antibody list is available as Supplementary Table 2. Cells were stained for 1 h at room temperature in the dark. After washing, samples were measured with a LSR II flow cytometer (BD). Representative gating strategies are shown in Supplementary Figs. 5–8.

### Cell sorting

Single-cell suspensions of splenocytes were enriched for CD4$^+$ T cells using a mouse CD4$^+$ T cell enrichment kit (Miltenyi Biotech, #130-104-454) according to the manufacturer's instructions. Enriched CD4$^+$ T cells and whole small intestine lamina propria lymphocytes were stained with anti-mouse CD45.1-Alexa Fluor 700, CD4-Brilliant Violet 711 and the non-fixable Live/Dead stain 7-Aminoactinomycin D

(Biolegend, #420403). Subsequently, live CD45.1$^+$ virus-specific SMARTA T lymphocytes per group were sorted on a FACS Aria II (BD) into tubes containing PBS with 10% FCS.

### Library preparation and single cell RNA sequencing

Gene expression libraries for single cell RNA libraries were prepared from sorted virus-specific SMARTA cells by using the 10x Genomics pipeline and the Chromium Next GEM Single Cell 3' Reagent Kits v3.1 according to the manufacture's user guide. Datasets related to single-cell RNA sequencing experiments that were generated and analyzed for the current study have been deposited and made publicly available in the Gene Expression Omnibus under the accession number GSE252214.

### Bioinformatic analysis of single cell RNA sequencing data

Read alignment and gene counting were performed with 10x Genomics Cell-Ranger v7.1. 0[89], using default parameters and pre-built mouse reference v2020-A (10x Genomics) based on mm10 GRCm38.p6 (release 98) and annotation from GENCODE Release M23. Downstream analysis was performed in R v4.3.0 with the R package Seurat v4.3.0[90]. Cells with >1000 genes, <10% mitochondrial genes, and UMI counts within the values of the 2$^{nd}$ and 98$^{th}$ quantiles were retained, along with genes detected in ≥ 3 cells. Filtered read counts from each sample were normalized independently with sctransform v0.3.5[91] using the glmGamPoi method[92] and *vst.flavor* v2. Integration features were identified from top 1000 highly variable genes excluding mitochondrial, ribosomal and TCR genes before calculation. Anchors between cells from different samples were identified on the integration features using reciprocal PCA and selected using the first 20 dimensions and two neighbors. Data integration was performed considering 50 neighbors to weight the anchors. PCA was calculated for the integrated data on the top 1000 highly variable genes. Both KNN graph and UMAP (*spread* 1, *min.distance* 0.3) were computed on the 30 nearest neighbors and first 20 PCA dimensions. Louvain clusters were identified using the Shared nearest neighbor (SNN) modularity optimization-based algorithm at resolution 0.5. Differential expression was performed using the Wilcoxon Rank Sum test and Bonferroni correction.

### Statistical analysis

General statistical analysis was performed with GraphPad Prism 5 software (GraphPad Software. Inc.). Student's $t$-test (two-tailed, unpaired) or one-way ANOVA using Tukey's multiple comparisons test were used as indicated in figure legends. A $p$ value < 0.05 was considered significant.

### Reporting summary

Further information on research design is available in the Nature Portfolio Reporting Summary linked to this article.

## Data availability

All data are included in the Supplementary Information or available from the authors, as are unique reagents used in this article. The raw numbers for charts and graphs are available in the Source Data file whenever possible. Raw datasets related to single-cell sequencing experiments of this study have been deposited and made publicly available in the Gene Expression Omnibus under the accession number GSE252214. Source data are provided with this paper.

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

## Acknowledgements

This research was supported by the Collaborative Research Center 1371 funded by the German Research Foundation (project number 395357507 P07, P14, Z01 and Z02). The Ohnmacht lab was further supported by grants from the European Research Council (ERC Starting grant, project number 716718) and the German Research Foundation (Project Number OH 282/1-2 within FOR2599, Project Number 490846870 – TRR355/1 TP05). Animal husbandry and experimental infrastructure were provided by the TUM Animal Research Center (ARC). We also acknowledge excellent technical support by the Comparative Experimental Pathology (CEP) at TUM.

## Author contributions

D.K. conducted all experiments with help from M.K., A.K., A.A. and A.v.S. C.W. G.P. helped with scRNAseq bioinformatic analysis. M.P. and K.N. performed 16S amplicon sequencing. S.B. and M.B. provided gnotobiotic and germfree animals. K.S. performed histological analysis. C.S.-W. and B.S. gave input and advice. C.O. supervised the study with help from D.Z. C.O. and D.K. wrote the manuscript with input from all authors. All authors read and approved the work.

## Funding

## Competing interests

The authors declare no competing interests.
