## [Transparent Peer Review file · Nature Communications]

A specific microbial consortium enhances Th1 immunity, improves LCMV viral clearance but aggravates LCMV disease pathology in mice

Corresponding Author: Dr Caspar Ohnmacht

Version 0:

Reviewer comments:

Reviewer #1

(Remarks to the Author)

This is a very timely article studying how a minimal microbiome consortium (OMM) can alter viral clearance and CD4 T cell responses in a mouse model of a chronic systemic LCMV viral infection. The authors evaluate alterations in T helper cell responses in OMM treated mice that have enhanced viral clearance and increased weight loss as compared to SPF and gnotobiotic mice. They thereby show how a defined and low-complexity microbial consortia can alter adaptive immune responses and possibly increase viral clearance at the cost of increased systemic inflammation. The studies are important to the field because the limited microbial consortia open avenues to identifying which specific bacteria or metabolites are effecting changes in T helper cell responses and are possible future drug/therapeutic targets.

Major comments

The authors's work is detailed, systematic, and support their conclusions, major comments suggested for revision are below.

The authors consistently describe weight loss as synonymous with increased immunopathology and they reference that the LCMV is non cytopathic. However, they do not show histologic changes (lymphocytic infiltrates) in the intestine or other organs supporting the hypothesis that increased inflammation was causing weight loss. Please better define immunopathology and use weight loss instead of immunopathology where appropriate across the manuscript text.

The authors attribute increased viral clearance to alterations in CD4 T cell subsets in OMM treated mice. However, as the authors note themselves in their discussion, alterations in innate immune responses may underlie changes in CD4 T cell subsets in the OMM treated mice. OMM treatment may alter Type I/III IFN baseline tone, innate immunity can be sanitizing, and could also be responsible for the increased viral clearance observed in OMM mice. This might help explain why there are no associated histologic changes in OMM treated mice who have enhanced viral clearance. Suggest that the authors characterize innate immune responses and viral titers in SPF, OMM and GF treated mice at baseline and early timepoints in order to address this possibility. This would address whether innate immune responses are confounding associations between viral clearance and alterations in CD4 T cell subsets.

Finally, suggest that the authors also include the limitations of utilizing a mouse model, a rodent-borne virus and a limited microbiota consortium in extrapolations to human diseases in the discussion.

Minor comments

General

Suggest that an English speaker reviews the text, particularly the abstract text
- Example, first sentence of abstract – does effective vaccination vary or does the intestinal microbiome vary?

Section 1

Do OMM treated mice without LCMV viral infection have stable weights over time?

Line 138 – please show data on CD8 T cells by study arm and over time in Supplementary Materials.

Line 181, "yet this effect could not be completely reversed by CD4 T cell depletion due to the high variation within individual groups", viral titer also appears quite homogenous in the spleen, suggest to remove

Line 182, "CD4 depletion increased the viral titer in SPF animals suggesting a role for CD4 T cells in viral control" show statistics supporting this statement (now not evident in Fig 2C)

Line 183-5, authors describe histology assessments for inflammation. Were these experiments performed for the non CD4 depleted mice? If so, please show to support argument that viral control phenotype is supported by increased intestinal and splenic inflammation.

Is Figure 2 G significant? If so please show, if not, please remove text "slight increase in intraepithelial lymphocytes OMM12 treated. Also hard to see sample sizes in 2E-G, scatter possible?"

Fig 2H-J, statistical representation here is a bit hard to read.

Line 197-201, suggest to move to discussion

Line 230 – what do the authors mean by "yet is resistant to infection driven outgrowth of individual bacteria."

Couldn't pathway be innate immunity => cd4 changes? Figure 1d, at 8dpi, were there also reduced viral loads at earlier time points suggesting alteration of innate immune responses

Line 348/line 355 - at the cost of enhanced immunopathology, suggest to be more specific by what is meant with immunopathology in absence of supporting histopathology.

Reviewer #2

(Remarks to the Author)

In this study Kolland et al describe how mice harboring a defined microbiome (OMM) show increased weight loss and viral control versus SPG or GF mice after LCMV Clone 13 infection. The authors successfully ascribed the enhanced weight loss to CD4 T cells and observed increased expansion of virus-specific CD4 T cells as well as preferential differentiation into Th1 cytotoxic phase, mostly in intestinal Lamina Propria. They did not however explain the mechanisms leading to the aforementioned CD4 T cell differences and how these differences lead to augmented immunopathology. In general, the study is mostly descriptive and while some findings are interesting there is lack of mechanistic data and some concerns, as mentioned below.

Below a list of major points:

1- The authors define that in the presence of OMM microbiome there is "severe immunopathology" however, the only effect described is related to higher weight loss (no changes in liver pathology or other tissues were observed). However, later during discussion they mention that mice harboring OMM microbiome succumb to pathology (396-399) but this is not shown in this paper. When do mice die? If they do die a survival graph is needed.

2- The authors do transfer experiments in figure 3 and see high influx of virus specific cells. This should also be extended to polyclonal virus-specific CD4 T cells using tetramer staining.

3- No mechanism is provided for how CD4 T cells (or which CD4 T cell subset) drive weight loss/immunopathology. For example, since CD4 T cells produce more TNF- α , is this pathology TNF- α -mediated?

4- The analysis of the single cell data may be affected by the low number of cells in GF mice. For example: *Cluster 3 is more abundant in the spleen of OMM than SPF, but concluding about GF is difficult since there are only few cells. Also, this cluster seems more abundant in the small intestine of OMM? (looking at UMAP) perhaps a violin plot to visualize this will help? *Cluster 6 is significantly higher in OMM but conclusions on GF are again hard to make because of the number of cells

5- The authors claim an increase in proliferative and cytotoxic CD4 T cells but no proliferation measurement or killing assay are shown

6- No experiments attempt to investigate which bacterial species or host factor drives the reported CD4 T cell changes in OMM mice

Minor comments for authors to consider

- CD4 depletion effect on viral titers are trending but not significant, but are referred as "crucial role".

- It is interesting that when virus-specific CD4 are transferred into mice the viral titer differences in the spleen are lost. Why do the authors think this is happening?

- Fig 3E the authors state that effect in viral titers was much less pronounced, the correct assertion is that there is no significant effect on the spleen.

- In Figure 4, While there are differences in the small intestine between SPF and OMM, they are not as dramatic perhaps as the change in spleen CD4 T cells. Which is interesting because the main changes so far have been observed in the small intestine. Why could this be?

- Missing ref page 382

- Line 437 should say "important" not "import"

- Need to add number of mice to fig legends or methods.

- In Lines 407 refers as how the authors didn't see any differences on CD8 T cell responses in this study, however no CD8 T cell data is shown here. The authors do refer to a previous study on bioRxiv from their group (lines 138-139) so they need to

refer to this study in line 407.

Reviewer #3

(Remarks to the Author)

In the manuscript "A specific intestinal microbial 1 consortium improves viral clearance by enhancing Th1 immunity after LCMV infection at the cost of immunopathology" Kolland D et al investigate the impact of a defined consortium of intestinal commensal bacteria (OMM-12) on the immune response to systemic LCMV infection. The authors observe increased weight loss in LCMV infected mice colonized with OMM-12 that is connected with a more robust LCMV-specific CD4 T cell response that both decreases viral burden but also drives immunopathology that results in weight loss. Single cell RNA-sequencing of LCMV-specific SMARTA transgenic CD4 T cells reveal OMM-12 skews this population to increased Th1 cell type in the intestine at the expense of more T_H phenotype in the spleen. The manuscript and data in figures are cleanly presented. The conclusion from the CD4 depletion experiments is confounded by the known impact of CD4 depletion on viral burden and CD8 exhaustion. Further the signals from OMM-12 that skew the CD4 T cell response are not explored. However this manuscript and results remain of high interest to the field.

Major Critiques

1. Figure 2 present data from an experiment that depletes CD4 T cells to test whether OMM-12 is working exclusively through CD4 T cells to lower viral burden. However, this experimental design is confound by the established effect of CD4 depletion on LCMV clone 13 infection. This system has been used to establish chronic, circulating viremia and pronounced CD8 T cell exhaustion. This has been reported in several papers such as (Barber et al Nature 2006; Matloubian et al 1994 J. Virology). The previous work using CD4 depletion in LCMV should be addressed in the manuscript. How do the author interpret their data in light of these previous works that show CD8 T cell function decreases follow CD4 depletion? Supplemental Figure 2 does not appear to observe CD8 T cell dysfunction following CD4 depletion. Further Figure 2C does not show increase viral burden in the spleen of anti CD4 OMM-12 mouse. Lack of CD4 help has been show to lead to elevated viral burden. Can authors elaborate why OMM-12 mice that received CD4 depletion would not experience elevated viral burden?

2. The diminished T_H response in OMM-12 response potential suggest an impaired humoral response. Can the authors demonstrate differences in the LCMV-specific antibody response to demonstrate a functional impact of the diminished T_H response?

Minor Critiques

1. Can the authors display the ICS flow data plots with Ifng and Tnfa in the same plot and then quantify the frequency and number of dual producing gp61 responsive CD4 T cells?

Version 1:

Reviewer comments:

Reviewer #1

(Remarks to the Author)

Many thanks to the authors for their extensive review of their manuscript, "Early bacterial microbiome is associated with positive vaccine take and shedding in neonatal schedule of the human neonatal rotavirus vaccine, RV3-BB." The reviewer appreciates the extensive responses to each of the raised comments, the manuscript is much improved due to the narrowed selection of relevant vaccine endpoints as well as time points.

Major comments

The reviewer (apologies) continues to have misgivings about the number of statistical tests in the manuscript and concomitant risk of multiple testing alongside a sometimes selective presentation of positive associations over negative associations.

The authors have kept three major vaccine outcomes – vaccine seroconversion, vaccine shedding, and a very large cumulative outcome of "vaccine take". However, they also have included sub outcomes in which every dose of vaccine shedding and take are included in the analysis. There does seem to be a grading in the quality of these associations, where

the likelihood of false statistical associations increases significantly as the authors move from seroconversion through shedding to take. The reviewer suggests that the authors consider restricting these outcomes to binary outcomes to mitigate this risk. For example, (assuming the same mechanism would be at play for each association) to not assess shedding separately by every time point, but make an outcome that is yes or no for shedding at any time point. The timepoint specific outcomes can be presented in supplementary materials. See specific suggestions in text.

Next, the reviewer suggests that the authors better build their rationale for their refined hypotheses in the introduction. What the literature is that supports this stated hypothesis. Why do the authors (mechanistically) think that microbiome alpha diversity and composition is associated with vaccine shedding and take for neonatal arms but not infants study arms. What are possible virologic or immunologic explanations that have led the authors to make this hypothesis?

Finally, the reviewer suggests that the authors address the possibility that better vaccine replication alters microbiome composition (infants' microbiome are more easily perturbed) and that associations between shedding and microbiome alpha diversity are a result of vaccine administration and not facilitating the vaccine's performance.

Minor comments (and expansions on major comments) by line item below

Abstract (line 44) suggest association instead of impact

Abstract (line 49) "compared to the placebo group, participants receiving three doses of RV3-bb in the neonatal schedule have higher alpha diversity" is unclear. Higher alpha diversity in baseline samples? Higher alpha diversity following vaccination?

Abstract (line 51) Suggest to define shedding, as not defined within the abstract

Abstract (line 53) suggest to modify to 'may present'

Include anti-RV IgA seroconversion results in the abstract

Introduction

Line 77, suggest to minimally add mode of delivery, breast milk, solid foods, antibiotic insults to development of the microbiome

Line 81, add "preterm neonates"

Line 95, remove "positive characteristics" or clearly cite literature that suggests that the outcomes listed (high alpha diversity, differences in beta diversity, and bacterial taxa profile) are "positive."

Line 100 "one of dose", typo?

Line 107 Suggest to add: and placebo recipients over time.

Line 112, suggest to alter “difference in beta diversity” to “beta diversity”

Line 114 “these data suggest that the early gut microbiome provides a gut environment that optimizes the potential for a positive vaccine response at that time point” this is quite speculative for an introduction, suggest to move to conclusion? Additionally, one could make the opposite argument – that only with the birthdose RV3-BB administration is there a significant association with microbiome composition, therefore other vaccines and timepoints are less dependent on microbiome background characteristics for vaccine protection. Could the authors comment?

Line 118, do the authors mean over time and following vaccination?

Line 119 “who did not shed the vaccine virus” please specify which schedule and dose this is relevant to.

Line 120 - A higher abundance of these bacteria might also suggest that these infants have had other insults (antibiotics/GE/hospital birth/relative immune compromise) that alter their microbiota and therefore also their immune response. Suggest to include this in considerations in conclusion. Further, decreased shedding at dose 2 might be reflective of induction of immunity at dose 1, therefore specify which dose these findings are relevant to.

Note that the authors are not presenting their anti-RV IgA seroconversion results in their introduction or abstract. The lack of association between IgA seroconversion and microbiome composition seems to me a very important finding in this study given that anti-RV IgA is the best available correlate of protection for rotavirus vaccines. The authors should lead with this information, include it clearly in their abstract, and discuss it’s relative importance clearly in their conclusion.

Authors’ rebuttal states:

Fig 1 describes the timepoints when the baseline stool was taken. In the Malawi study, a baseline stool was taken soon after birth prior to administration of the first dose of vaccine or placebo. In the Indonesia study the first stool was collected in the first week of life – 3-7 days after administration of the first dose of vaccine or placebo.

Suggest to include this before result reporting to facilitate interpretation of country-specific results.

Results

Line 170, clarify how the stool samples for these time points were selected – was just any stool sample taken, or the closest to vaccination? For these timepoints, the microbiome is likely altered by rotavirus vaccine strain replication. Therefore associations between vaccine shedding and microbiome are logical, given that the highest vaccine replication will likely perturb the microbiome the most. Please rebut or include this consideration in your discussion.

Vaccine response in association with alpha and beta diversity

Very strong suggestion to include the IgA seroconversion response in the main text alongside the stool shedding and positive vaccine take, given that these are endpoints defined by the authors and that the IgA seroconversion endpoint has the best evidence that it correlates with vaccine protection.

Table 2/Fig2

I appreciate that the authors took the time to collate their study results into this table and figure, but it remains complex to the reviewer, and therefore will likely be complex to the reader. It appears that there are separate analyses for whether or not there was vaccine take or shedding following 1, 2, or 4 doses for both schedule groups. See General comments. Could the authors further simplify by simplifying their endpoints. This reduces the number of statistical analyses, increases the understandability and improves comparability across country and schedule groups.

Suggest that the authors follow the approach they describe in their rebuttal: divide children in two 3 groups per study arm

and simplify: ever had vaccine take (Y/N), ever had stool shedding (Y/N), seroconversion (Y/N), presented by study arm. the understandability and improves the comparability across study arms of the study results. This table and fig2 could be moved to the supplementary.

Line 251 – suggest to not only describe the positive findings and equally note the lack of separation at other doses (and timepoints)? This framing suggests that there are important microbiome differences, but the preponderance of the evidence is actually not showing differences by vaccine endpoints for microbiome composition, which seems just as valuable an outcome to report.

Same suggestion for simplification in Figures 3 and 4, with per dose outcomes in supplementary

Figure 4, if vaccine take definition did not include shedding, was the difference in PCoA at 6 weeks maintained?

Line 357, consider replacing PCA directionality with age directionality for taxa (like Bacteroides, Escherichia and Streptococcus) that distinguish separation across the axes

Bacterial taxa influence RV3-BB vaccine response

Line 379, suggest to change subtitle to “Bacterial taxa associate with RV3-BB vaccine response”

Line 385, confirm no differences by seroconversion

Line 460 – were there differences in breastfeeding rates between the two study cohorts given Bifidobacterium difference

Paragraph 453, is the reviewer correct that there was no re-analysis of associations between microbiome and Rotarix study vaccine endpoints in Malawi, only a comparison across ages for all included infants, regardless of vaccine take, shedding and seroconversion? This is confusing given data prior that evaluates associations with vaccine performance. Suggest to clarify in text.

Discussion

Line 520, what is meant by “shedding following”, this sentence is somewhat confusing, suggest to rephrase.

Line 523, please discuss that increased microbiome alpha diversity associating with shedding may be confounded by effective shedding increasing alpha diversity.

Paragraph 546, suggest to name the vaccine studied for the cited literature for clarity.

Reviewer #2

(Remarks to the Author)

The authors have satisfactorily addressed all my concerns

Reviewer #3

(Remarks to the Author)

Thank you for your detailed response to all of the Reviewer comments. Thank you for reinforcing in your comments that this manuscript is looking at the acute phase of CI-13 infection (day 4,8) due to the OMM12 mice losing weight that reaches the humane endpoint of the study. This was stated by the authors in the original and revised manuscript (Ln 126) but was missed/misinterpreted by this Reviewer (please make a spelling correction “human” to “humane” on Ln 126). This is a stark observation that 100% (or at least the majority) of OMM12 mice needed to be euthanized due to weight loss while 0% of the other groups did. It strengthens the important phenotypic difference observed between the SPF /GF and OMM12 mice. Please include a graph visually showing this data to the reader and detail what the humane endpoint criteria was for your

study. Similar to a survival graph but with time when the humane endpoint was reached. This information is important to appreciate the extent of disease pathology reached with the OMM12 mice and also valuable data for future researchers to have to provide rationale to adjust their humane endpoint criteria as needed.

1. Thank you for the Authors response to this point and included several references on the impact of CD4 depletion in LCMV CI-13 infection. We agree these studies focus on the loss of function of CD8 T cells at chronic timepoints after infection but this manuscript could not investigate this later timepoint due to the OMM12 mice reaching humane endpoint.

The viral titers in Figure 2C demonstrate that CD4 depleted OMM12 mice do not observe an increase in viral titers compared to OMM12 mice with CD4 T cells. Since the goal of the experiment in Figure 2 was to assess if the microbiome (OMM12) acted through CD4 T cells to alter disease pathology and viral clearance (Ln 157-158), the authors should state in this section that while the CD4 T cells are driving weight loss (disease pathology) in OMM12 CI-13 infected mice, CD4 T cells are not the causative agent that drive diminished viral titers in OMM12 mice.

2. Thank you for the Authors response to this point and highlighting again that OMM12 mice reach a humane endpoint early and thus the humoral response later in the infection can not be assessed.

3. Thank you for adjusting the FACS plots in Figure 1F to show IFN γ /TNF α . A point of clarity, this Reviewer was asking for the total number of IFN γ /TNF α double positive cells from Fig 1F to be graphed out. Similar to the quantification of absolute number of Th1 CD4 cells in Fig 1E.

Version 2:

Reviewer comments:

Reviewer #1

(Remarks to the Author)

The authors have satisfactorily addressed all my concerns. My thanks for all their detailed answers.

Reviewer #3

(Remarks to the Author)

In reference to Point 1:

Please include the humane endpoint criteria as defined by animal welfare criteria that was used in this study. Future studies and investigator will great benefit from this data as rationale to seek alternative metrics for a humane endpoint.

In reference to Point 3:

While efficiency of cell recovery from lamina propria prep is a technical challenge, the authors were able to display absolute number of T-bet $^+$ cells from the small intestine in Fig. 1E. It is inconsistent to display absolute numbers of cells in Fig 1E but not Fig 1G since both pieces of data required lamina propria isolation. This Reviewer suggests display the data from both as relative absolute number. For Fig 1E this would be Tbet $^+$ CD4 T cells per 1×10^6 CD45 $^+$ cells and Fig 1G as cytokine $^+$ CD4 T cells per 1×10^6 CD45 $^+$ cells. These readouts are still relative numbers but takes into account relative abundance of parent CD4 T cell populations.

REVIEWER COMMENTS

We thank the reviewers for their time and thorough review of our manuscript that substantially helped to improve and strengthen the clarity of our manuscript.

We do understand that this study opens up many questions, e.g. which bacterial component from which strain is responsive for the effect described in the present manuscript or whether innate immune cells act as accessory cells for the effect of the OMM¹² consortium to shape anti-viral T helper cell responses. These are very important questions, but we feel they are better addressed in a separate study because they require a number of sophisticated additional experiments (and respective control experiments) that are technically challenging and very time-consuming due to the necessity to perform infections under gnotobiotic conditions. Likewise, it is not simply possible to use knockout- or reporter animals because these strains are not available under germfree/OMM¹² conditions. Finally, we explicitly focused in this manuscript on virus-specific T helper cell responses and not on the innate immune system as previous reports focused already on the general effect of a complex microbiota on innate immune cells.

A detailed color-coded pt-to-pt reply is provided below.

Reviewer #1 (Remarks to the Author):

This is a very timely article studying how a minimal microbiome consortium (OMM) can alter viral clearance and CD4 T cell responses in a mouse model of a chronic systemic LCMV viral infection. The authors evaluate alterations in T helper cell responses in OMM treated mice that have enhanced viral clearance and increased weight loss as compared to SPF and gnotobiotic mice. They thereby show how a defined and low-complexity microbial consortia can alter adaptive immune responses and possibly increase viral clearance at the cost of increased systemic inflammation. The studies are important to the field because the limited microbial consortia open avenues to identifying which specific bacteria or metabolites are effecting changes in T helper cell responses and are possible future drug/therapeutic targets.

We thank the reviewer for acknowledging the timeliness and the importance of our study.

Major comments

The authors's work is detailed, systematic, and support their conclusions, major comments suggested for revision are below.

We thank the reviewer for also this positive evaluation.

The authors consistently describe weight loss as synonymous with increased immunopathology and they reference that the LCMV is non cytopathic. However, they do not show histologic changes (lymphocytic infiltrates) in the intestine or other organs supporting the hypothesis that increased inflammation was causing weight loss. Please better define immunopathology and use weight loss instead of immunopathology where appropriate across the manuscript text.

We thank the reviewer for this comment that was also raised by reviewer 2. In fact, the word immunopathology is not always clearly defined, and we have now changed this to the term 'disease pathology'. We would also like to mention that the levels of pathology we look at here are difficult to grasp with classical histological methods.

As LCMV CI-13 is a non-cytopathic virus, all disease pathology is based on the activity of immune cells. We want to strengthen that we show throughout the manuscript enhanced secretion of effector cytokines (IFN-g and TNF-a) from CD4⁺ T cells and - based on transcriptional profiles - some indication for enhanced capacity of cytotoxicity by these cells which may directly cause immune- or

disease pathology by various pathways. It has been shown previously that vaccine-elicited CD4 T cells are capable to induce a fatal immunopathology after infection with LCMV CI-13 (PMID: 25593185). At the early time points investigated here (8 dpi) we did not find any evidence for inflammatory cell infiltration in the intestinal tract, but it remains formally possible that this is the case for some other organs.

The authors attribute increased viral clearance to alterations in CD4 T cell subsets in OMM treated mice. However, as the authors note themselves in their discussion, alterations in innate immune responses may underlie changes in CD4 T cell subsets in the OMM treated mice. OMM treatment may alter Type I/III IFN baseline tone, innate immunity can be sanitizing, and could also be responsible for the increased viral clearance observed in OMM mice. This might help explain why there are no associated histologic changes in OMM treated mice who have enhanced viral clearance. Suggest that the authors characterize innate immune responses and viral titers in SPF, OMM and GF treated mice at baseline and early timepoints in order to address this possibility. This would address whether innate immune responses are confounding associations between viral clearance and alterations in CD4 T cell subsets.

Indeed, innate immune cells are one likely cell type that may be influenced by the OMM¹² consortium. The general impact of commensal microbes on innate anti-viral immunity has been shown before (e.g. PMID: 22705104, PMID: 32380006) but whether and how a defined minimal community in general affects innate immune cells will require a detailed analysis to identify a) the relevant innate immune cell type/subset, b) the key bacterial strain or metabolite mediating such an effect and c) the mode of action/the affected pathway targeted by the bacterial metabolite. All of these investigations require elaborate methodologies such as derivatives of the OMM¹² consortium, transcriptional profiling of all kinds of innate immune cells (DC subsets, macrophages, monocytes, innate lymphoid cells including NK cells) and validation of such putative effects on protein level and knockout of the respective targeted pathway in innate immune cell for validation of the effect under SPF conditions and artificial treatment with bacterial metabolite(s)/colonization with bacteria lacking the capacity to generate such metabolites. Given the high methodical, technical and logistic hurdles we feel this is not feasible in a reasonable amount of time and warrants a separate study.

Finally, suggest that the authors also include the limitations of utilizing a mouse model, a rodent-borne virus and a limited microbiota consortium in extrapolations to human diseases in the discussion. A respective section has been included at the end of the discussion.

Minor comments

General

Suggest that an English speaker reviews the text, particularly the abstract text
- Example, first sentence of abstract – does effective vaccination vary or does the intestinal microbiome vary?

The English has been corrected throughout the text by a native English speaker.

Section 1

Do OMM treated mice without LCMV viral infection have stable weights over time?

Yes, this has already been shown by others (PMID: 34795236 and PMID: 36126044) and the references are included now in the result section.

Line 138 – please show data on CD8 T cells by study arm and over time in Supplementary Materials. The impact of CD8 T cells has been investigated in a separate study (currently available at BioRxiv under doi.org/10.1101/2022.10.03.510696, see also Supplementary Figures in this article). Since we didn't observe any obvious effect in this study and the results presented in Supplementary Figure 2 of this study did not indicate a profound effect of the microbiome at the time point investigated here, CD8 T cells have not been further studied in this study.

Line 181, “yet this effect could not be completely reversed by CD4 T cell depletion due to the high

variation within individual groups “, viral titer also appears quite homogenous in the spleen, suggest to remove.

Removed.

Line 182, “CD4 depletion increased the viral titer in SPF animals suggesting a role for CD4 T cells in viral control” show statistics supporting this statement (now not evident in Fig 2C)

This has been rephrased more carefully and a p value has been added.

Line 183-5, authors describe histology assessments for inflammation. Were these experiments performed for the non CD4 depleted mice? If so, please show to support argument that viral control phenotype is supported by increased intestinal and splenic inflammation.

Histological assessments have only been done for small intestine as the site of colonization with the OMM¹² consortium but also as the site with the highest expansion of virus-specific T cells (Fig. 1 and 3). The histology shown in Fig. 2D has been done for both isotype and CD4-depleted sections of the small intestine as indicated in the text and Figure legend.

Is Figure 2 G significant? If so please show, if not, please remove text “slight increase in intraepithelial lymphocytes OMM12 treated. Also hard to see sample sizes in 2E-G, scatter possible?

The sentence has been removed as this was only a tendency and not significant. The sample size is now given also in the legend to improve clarity.

Fig 2H-J, statistical representation here is a bit hard to read.

Statistical representation has been improved.

Line 197-201, suggest to move to discussion

Has been moved to discussion.

Line 230 – what do the authors mean by “yet is resistant to infection driven outgrowth of individual bacteria. “

Sentence has been re-phrased to make clear that the infection-induced inflammation per se did not result in a fundamental alteration in the composition of the OMM¹² community, e.g. the selective expansion or reduction in the relative abundance of one bacterial strain as a consequence of inflammation/ongoing anti-viral immune response.

Couldn't pathway be innate immunity => cd4 changes? Figure 1d, at 8dpi, were there also reduced viral loads at earlier time points suggesting alteration of innate immune responses

Indeed, innate immune cells are one likely accessory cell type that may be influenced by the OMM¹² consortium. This question will also be part of future studies, but the focus of this study was on a potential impact of the OMM¹² on adaptive immunity and particularly on CD4⁺ T helper cells.

Line 348/line 355 - at the cost of enhanced immunopathology, suggest to be more specific by what is meant with immunopathology in absence of supporting histopathology.

As outlined above the term ‘immunopathology’ has been replaced by ‘disease pathology’ and an explanation has been now given.

Reviewer #2 (Remarks to the Author):

In this study Kolland et al describe how mice harboring a defined microbiome (OMM) show increased weight loss and viral control versus SPG or GF mice after LCMV Clone 13 infection. The authors successfully ascribed the enhanced weight loss to CD4 T cells and observed increased expansion of virus-specific CD4 T cells as well as preferential differentiation into Th1 cytotoxic phase, mostly in intestinal Lamina Propria. They did not however explain the mechanisms leading to the aforementioned CD4 T cell differences and how these differences lead to augmented immunopathology. In general, the study is mostly descriptive and while some findings are interesting there is lack of mechanistic data and some concerns, as mentioned below.

Below a list of major points:

1- The authors define that in the presence of OMM microbiome there is “severe immunopathology” however, the only effect described is related to higher weight loss (no changes in liver pathology or other tissues were observed). However, later during discussion they mention that mice harboring OMM microbiome succumb to pathology (396-399) but this is not shown in this paper. When do mice die? If they do die a survival graph is needed.

As outlined in the response to reviewer 1, we have now changed the term ‘immunopathology’ to ‘disease pathology’ throughout the manuscript. Enhanced secretion of effector cytokines (IFN- γ and TNF- α) from CD4⁺ T cells and the capacity for cytotoxicity may directly cause disease pathology without inflammatory cell infiltration. Importantly, it has been shown previously that vaccine-elicited CD4 T cells are in principle capable to induce a fatal immunopathology after infection with LCMV Cl-13 (PMID: 25593185). This has now been included in the discussion. In the referred paper this was associated to immunopathology and inflammatory cell infiltration/tissue damage, but the authors there used a vaccination model to induce LCMV-specific CD4⁺ T cells which is not comparable to the question addressed here.

As outlined throughout the manuscript, the experiments had to be stopped at day 8 post infection because most animals of the OMM¹² group reached human endpoint criteria due to massive body weight loss – this is also mentioned in the results and discussion. Therefore, no survival graph can be shown.

2- The authors do transfer experiments in figure 3 and see high influx of virus specific cells. This should also be extended to polyclonal virus-specific CD4 T cells using tetramer staining.

We want to point out that our restimulation experiments (Figure 1 and 3) have been performed all with the H-2^b- immunodominant peptide gp61 of LCMV and thus describe the pool of endogenous virus-specific CD4 T cells capable of secreting effector cytokines indicating that also the endogenous T helper cell repertoire is affected in a similar manner by the OMM¹² consortium.

Furthermore, in our hands, tetramer staining has shown technical limitations in other projects which is why we refrained from using this technology for this study and rather relied on adoptive transfer of SMARTA cells and gp61-restimulation experiments.

3- No mechanism is provided for how CD4 T cells (or which CD4 T cell subset) drive weight loss/immunopathology. For example, since CD4 T cells produce more TNF- α , is this pathology TNF- α -mediated?

This is an important point but there may not be only one answer to this question. It is possible that TNF- α contributes to disease pathology as TNF- α has been linked to excessive inflammation in COVID-19 or autoimmune diseases (PMID: 35348444, PMID: 33800290). Other cytokines like IFN- γ or the capacity for cell cytotoxicity could play a similar role or a combination of all of these factors. A definitive answer or even only exclusion for one of these factors would require the use of cytokine- or cytotoxicity-deficient mice under gnotobiotic conditions which is currently not realistic given the technical and time-/capacity hurdles for gnotobiotic experiments. The sole transfer of SMARTA cells deficient for TNF- α (or any other inflammatory factor) would not address this question because also animals without adoptively transferred SMARTA cells show weight loss/disease pathology (Figure 1). Moreover, attempts to neutralize cytokines in vivo are difficult to control and will inadvertently target all cell types and not solely CD4 T cells.

4- The analysis of the single cell data may be affected by the low number of cells in GF mice. For example: *Cluster 3 is more abundant in the spleen of OMM than SPF, but concluding about GF is difficult since there are only few cells. Also, this cluster seems more abundant in the small intestine of OMM? (looking at UMAP) perhaps a violin plot to visualize this will help? *Cluster 6 is significantly higher in OMM but conclusions on GF are again hard to make because of the number of cells. Indeed, due to technical difficulties the cell number of re-isolated SMARTA cells from germfree animals is quite low. We have also considered to remove the germfree condition from the analysis for this reason but feel it is useful for readers as a comparison. We have modified the text accordingly in the description and also base our analysis primarily on the comparison OMM¹² – SPF conditions.

For Figure 4A, we feel UMAPs are useful to visualize all clusters and the relative quantification of clusters for each condition is depicted as bar graphs in Fig. 4B.

Violin plots have been included to better visualize expression of Tfh-related genes across all clusters and genotypes in Figure 5A. However, the UMAPs represented better the expression for genes described in Figure 5D-G, which is why they were chosen for the official Figures.

Please find Violin plots for some genes presented in Fig. 5 and Supplementary Figure 4 below:

Violin plots of genes related to Fig. 5:

Violin plots of genes related to Supplementary Fig. 4:

5- The authors claim an increase in proliferative and cytotoxic CD4 T cells but no proliferation measurement or killing assay are shown

The isolation of a cytotoxic CD4 T cell cluster is currently not possible to test directly as clear markers for their isolation are missing. We modified the text to make clear that no direct evidence for cytotoxicity is shown and it is only an interpretation from the transcriptional signature.

As we transferred always the same SMARTA cell number prior infection and then found enhanced SMARTA cell numbers in OMM¹² animals (especially in the small intestine), we indeed claim this is the result of enhanced proliferation. Any other explanation such as better survival of cells or impaired contraction phase is unlikely as the time point of analysis was in the acute phase of the infection.

6- No experiments attempt to investigate which bacterial specie or host factor drives the reported CD4 T cell changes in OMM mice

As outlined in the response to reviewer 1, this will be part of future investigations. Noteworthy, the bacterial OMM¹² community is quite stable over the course of the infection (see Figure 2H-J and Supplementary Figure 2&3) suggesting that the underlying effect may be present early on or even at steady state but requires a stimulus to manifest in alterations of CD4 T cell differentiation (e.g. LCMV infection). The identification of the bacterial species or respective metabolite will require the use of OMM¹² derivatives (by selectively removing or adding individual bacteria) in combination with LCMV infections. Some OMM¹² derivatives already exist (PMID: 36208631, PMID: 34857933 and PMID: 37553336) and can be used in the future but whether they will be useful to identify the bacterial species or metabolite remains to be explored. Other communities will have to be developed and first tested for their community stability over time before using them for infectious studies. The identification of host factors, e.g. effects within the innate immune system, have been elaborated in response to Reviewer 1.

Minor comments for authors to consider

- CD4 depletion effect on viral titers are trending but not significant, but are referred as “crucial role”. This has been rephrased.

- It is Interesting that when virus-specific CD4 are transferred into mice the viral titer differences in the spleen are lost. Why do the authors think this is happening?
Indeed, this is an interesting observation for which we currently don't have a direct explanation. One possibility may be that by enhancing the frequency of virus-specific T helper cells the decisive impact of the OMM¹² consortium on early T helper cell accumulation/gain of effector cell function in the spleen is overridden by increasing the number of naïve virus-specific T helper cells. Alternatively, the enhanced frequency of virus-specific T helper cells may override impact of the OMM¹² consortium on other cells involved in virus control such as NK cells, etc.

- Fig 3E the authors state that effect in viral titers was much less pronounced, the correct assertion is that there is no significant effect on the spleen.
Wording has been corrected.

- In Figure 4, While there are differences in the small intestine between SPF and OMM, they are not as dramatic perhaps as the change in spleen CD4 T cells. Which is interesting because the main changes so far have been observed in the small intestine. Why could this be?
It is possible that in the spleen (in which we did not observed differences in the accumulation of SMARTA cells), the OMM¹² consortium primarily affects cluster distribution (Tfh-Th1 effector differentiation) while in the small intestine, we observe an expansion of SMARTA cells under OMM¹² conditions that equally affects all clusters. As clusters from scRNAseq data do not reflect absolute numbers but only relative distribution of cells there may be a stronger effect on T helper cell clusters in the spleen and only to a lesser degree in the small intestine.

- Missing ref page 382

- Line 437 should say “important” not “import”
Corrected.

- Need to add number of mice to fig legends or methods.
Number of mice has been added to figure legends.

- In Lines 407 refers as how the authors didn't see any differences on CD8 T cell responses in this study, however no CD8 T cell data is shown here. The authors do refer to a previous study on biorxiv

from their group (lines 138-139) so they need to refer to this study in line 407.
The reference has been added.

Reviewer #3 (Remarks to the Author):

In the manuscript “A specific intestinal microbial 1 consortium improves viral clearance by enhancing Th1 immunity after LCMV infection at the cost of immunopathology” Kolland D et al investigate the impact of a defined consortium of intestinal commensal bacteria (OMM-12) on the immune response to systemic LCMV infection. The authors observe increased weight loss in LCMV infected mice colonized with OMM-12 that is connected with a more robust LCMV-specific CD4 T cell response that both decreases viral burden but also drives immunopathology that results in weight loss. Single cell RNA-sequencing of LCMV-specific SMARTA transgenic CD4 T cells reveal OMM-12 skews this population to increased Th1 cell type in the intestine at the expense of more Tfh phenotype in the spleen. The manuscript and data in figures are cleanly presented. The conclusion from the CD4 depletion experiments is confounded by the known impact of CD4 depletion on viral burden and CD8 exhaustion. Further the signals from OMM-12 that skew the CD4 t cell response are not explored. However this manuscript and results remain of high interest to the field.

We thank the reviewer for this positive evaluation.

Major Critiques

1. Figure 2 present data from an experiment that depletes CD4 T cells to test whether OMM-12 is working exclusively through CD4 T cells to lower viral burden. However, this experimental design is confound by the established effect of CD4 depletion on LCMV clone 13 infection. This system has been used to establish chronic, circulating viremia and pronounced CD8 T cell exhaustion. This has been reported in several papers such as (Barber et al Nature 2006; Matloubian et al 1994 J. Virology). The previous work using CD4 depletion in LCMV should be addressed in the manuscript. How do the author interpret their data in light of these previous works that show CD8 T cell function decreases follow CD4 depletion? Supplemental Figure 2 does not appear to observe CD8 T cell dysfunction following CD4 depletion.

Former studies have indeed shown that CD4 T cell depletion influences CD8 T cell functionality and exhaustion phenotypes but solely during late/chronic phases of the infection or recall infection (PMID: 7966595; PMID: 3153075; PMID: 31810883; PMID: 9858507; PMID: 12690201; PMID: 15300249; PMID: 12594515). The mentioned references as well as a few others have now been included in the text but we are not aware of any report describing such an effect during the acute phase of LCMV CI-13 infection. Thus, based on literature, we did not expect any impact of CD4 T cell depletion on CD8 T cells at the early time point investigated here (as indicated also in Supplementary Figure 2) but only during later stages of LCMV CI-13 infection. However, we were unable to perform experiments at later time points because OMM¹² animals lost substantially more body weight and reached human endpoint before.

Further Figure 2C does not show increase viral burden in the spleen of anti CD4 OMM-12 mouse. Lack of CD4 help has been show to lead to elevated viral burden. Can authors elaborate why OMM-12 mice that received CD4 depletion would not experience elevated viral burden?

We observed only a slight trend for elevated viral titers in CD4 T cell-depleted SPF and OMM¹² animals. However, as outlined above, CD4 T cell-mediated regulation of CD8 T cell functionality may only play a role at later stages of the infection that we were unable to study due to enhanced disease pathology. The slight trend for higher viral titers in CD4 T cell-depleted animals at this early time point may be derived from cytotoxicity of virus-specific CD4 T cells. Still, CD8 T cells may play by far a more important role for viral control which is why these effects may only be marginal.

2. The diminished Tfh response in OMM-12 response potential suggest an impaired humoral response. Can the authors demonstrate differences in the LCMV-specific antibody response to demonstrate a functional impact of the diminished Tfh response?

The LCMV-specific humoral immune response is not detected/doesn't play a significant role at the early time points post infection investigated in this study but only during the chronic phase of LCMV CI-13 infection (PMID: 31995746; PMID: 29196449). Due to the massive weight loss of OMM¹² animals we were unable to perform any experiments beyond day 8 after infection and measurements of LCMV-specific antibodies are therefore not useful in this regard. Nevertheless, our data indicate that also the early Tfh cell response is skewed by the OMM¹² bacterial consortium and may affect humoral response during chronic infection; yet other mechanisms contribute to excessive weight loss/disease pathology.

Minor Critiques

1. Can the authors display the ICS flow data plots with Ifng and Tnfa in the same plot and then quantify the frequency and number of dual producing gp61 responsive CD4 T cells?

Respective FACS plots and quantification are now shown in Fig 1F and G.

For Figure 3, a subdividing of SMARTA cells into 'double-producing' cells was not useful due to the low cell numbers of re-isolated SMARTA cells from the lamina propria of the SI.

PD Dr. Caspar Ohnmacht

ZAUM
Center of Allergy & Environment
Biedersteiner Str. 29
80802 Munich
Germany

T + 49 (0) 89 41 40 34 51
F + 49 (0) 89 41 40 34 52

<http://www.zaum-online.de>
zaum@tum.de

Pt-to-Pt reply revised manuscript

Date: 29th of January 2024

Name of corresponding author: Caspar Ohnmacht

Title: A specific microbial consortium improves viral clearance by enhancing Th1 immunity after LCMV infection at the cost of disease pathology

Manuscript number: NCOMMS-24-08228A

REVIEWER COMMENTS

A detailed color-coded pt-to-pt reply is provided below.

Reviewer #1 (Remarks to the Author):

Unfortunately, all the comments of Reviewer #1 do not refer to our but to another manuscript and we can therefore not address them.

Many thanks to the authors for their extensive review of their manuscript, “Early bacterial microbiome is associated with positive vaccine take and shedding in neonatal schedule of the human neonatal rotavirus vaccine, RV3-BB.” The reviewer appreciates the extensive responses to each of the raised comments, the manuscript is much improved due to the narrowed selection of relevant vaccine endpoints as well as time points.

Major comments

The reviewer (apologies) continues to have misgivings about the number of statistical tests in the manuscript and concomitant risk of multiple testing alongside a sometimes selective presentation of positive associations over negative associations.

The authors have kept three major vaccine outcomes – vaccine seroconversion, vaccine shedding, and a very large cumulative outcome of “vaccine take”. However, they also have included sub outcomes in which every dose of vaccine shedding and take are included in the analysis. There does seem to be a grading into the quality of these associations, where the likelihood of false statistical associations increases significantly as the authors move from seroconversion through shedding to take. The reviewer suggests that the authors consider restricting these outcomes to binary outcomes to mitigate this risk. For example, (assuming the same mechanism would be at play for each association) to not assess shedding separately by every time point, but make an outcome that is yes or no for shedding at any time point. The timepoint specific outcomes can be

presented in supplementary materials. See specific suggestions in text.

Next, the reviewer suggests that the authors better build their rationale for their refined hypotheses in the introduction. What the literature is that supports this stated hypothesis. Why do the authors (mechanistically) think that microbiome alpha diversity and composition is associated with vaccine shedding and take for neonatal arms but not infants study arms. What are possible virologic or immunologic explanations that have led the authors to make this hypothesis?

Finally, the reviewer suggests that the authors address the possibility that better vaccine replication alters microbiome composition (infants' microbiome are more easily perturbed) and that associations between shedding and microbiome alpha diversity are a result of vaccine administration and not facilitating the vaccine's performance.

Minor comments (and expansions on major comments) by line item below

Abstract (line 44) suggest association instead of impact

Abstract (line 49) "compared to the placebo group, participants receiving three doses of RV3-bb in the neonatal schedule have higher alpha diversity" is unclear. Higher alpha diversity in baseline samples? Higher alpha diversity following vaccination?

Abstract (line 51) Suggest to define shedding, as not defined within the abstract

Abstract (line 53) suggest to modify to 'may present'

Include anti-RV IgA seroconversion results in the abstract

Introduction

Line 77, suggest to minimally add mode of delivery, breast milk, solid foods, antibiotic insults to development of the microbiome

Line 81, add "preterm neonates"

Line 95, remove "positive characteristics" or clearly cite literature that suggests that the outcomes listed (high alpha diversity, differences in beta diversity, and bacterial taxa profile) are "positive."

Line 100 "one of dose", typo?

Line 107 Suggest to add: and placebo recipients over time.

Line 112, suggest to alter "difference in beta diversity" to "beta diversity"

Line 114 "these data suggest that the early gut microbiome provides a gut environment that optimizes the potential for a positive vaccine response at that time point" this is quite speculative for an introduction, suggest to move to conclusion? Additionally, one could make the opposite argument – that only with the birthdose RV3-BB administration is there a significant association with microbiome composition, therefore other vaccines and timepoints are less dependent on microbiome background characteristics for vaccine protection. Could the authors comment?

Line 118, do the authors mean over time and following vaccination?

Line 119 “who did not shed the vaccine virus” please specify which schedule and dose this is relevant to.

Line 120 - A higher abundance of these bacteria might also suggest that these infants have had other insults (antibiotics/GE/hospital birth/relative immune compromise) that alter their microbiota and therefore also their immune response. Suggest to include this in considerations in conclusion. Further, decreased shedding at dose 2 might be reflective of induction of immunity at dose 1, therefore specify which dose these findings are relevant to.

Note that the authors are not presenting their anti-RV IgA seroconversion results in their introduction or abstract. The lack of association between IgA seroconversion and microbiome composition seems to me a very important finding in this study given that anti-RV IgA is the best available correlate of protection for rotavirus vaccines. The authors should lead with this information, include it clearly in their abstract, and discuss it's relative importance clearly in their conclusion.

Authors' rebuttal states:

Fig 1 describes the timepoints when the baseline stool was taken. In the Malawi study, a baseline stool was taken soon after birth prior to administration of the first dose of vaccine or placebo. In the Indonesia study the first stool was collected in the first week of life – 3-7 days after administration of the first dose of vaccine or placebo.

Suggest to include this before result reporting to facilitate interpretation of country-specific results.

Results

Line 170, clarify how the stool samples for these time points were selected – was just any stool sample taken, or the closest to vaccination? For these timepoints, the microbiome is likely altered by rotavirus vaccine strain replication. Therefore associations between vaccine shedding and microbiome are logical, given that the highest vaccine replication will likely perturb the microbiome the most. Please rebut or include this consideration in your discussion.

Vaccine response in association with alpha and beta diversity

Very strong suggestion to include the IgA seroconversion response in the main text alongside the stool shedding and positive vaccine take, given that these are endpoints defined by the authors and that the IgA seroconversion endpoint has the best evidence that it correlates with vaccine protection.

Table 2/Fig2

I appreciate that the authors took the time to collate their study results into this table and figure, but it remains complex to the reviewer, and therefore will likely be complex to the reader. It appears that there are separate analyses for whether or not there was vaccine take or shedding following 1, 2, or 4 doses for both schedule groups. See General comments. Could the authors further simplify by simplifying their endpoints. This reduces the number of statistical analyses,

increases the understandability and improves comparability across country and schedule groups.

Suggest that the authors follow the approach they describe in their rebuttal: divide children in two 3 groups per study arm and simplify: ever had vaccine take (Y/N), ever had stool shedding (Y/N), seroconversion (Y/N), presented by study arm. the understandability and improves the comparability across study arms of the study results. This table and fig2 could be moved to the supplementary.

Line 251 – suggest to not only describe the positive findings and equally note the lack of separation at other doses (and timepoints)? This framing suggests that there are important microbiome differences, but the preponderance of the evidence is actually not showing differences by vaccine endpoints for microbiome composition, which seems just as valuable an outcome to report.

Same suggestion for simplification in Figures 3 and 4, with per dose outcomes in supplementary

Figure 4, if vaccine take definition did not include shedding, was the difference in PCoA at 6 weeks maintained?

Line 357, consider replacing PCA directionality with age directionality for taxa (like Bacteroides, Escherichia and Streptococcus) that distinguish separation across the axes

Bacterial taxa influence RV3-BB vaccine response

Line 379, suggest to change subtitle to “Bacterial taxa associate with RV3-BB vaccine response”

Line 385, confirm no differences by seroconversion

Line 460 – were there differences in breastfeeding rates between the two study cohorts given Bifidobacterium difference

Paragraph 453, is the reviewer correct that there was no re-analysis of associations between microbiome and Rotarix study vaccine endpoints in Malawi, only a comparison across ages for all included infants, regardless of vaccine take, shedding and seroconversion? This is confusing given data prior that evaluates associations with vaccine performance. Suggest to clarify in text.

Discussion

Line 520, what is meant by “shedding following”, this sentence is somewhat confusing, suggest to rephrase.

Line 523, please discuss that increased microbiome alpha diversity associating with shedding may be confounded by effective shedding increasing alpha diversity.

Paragraph 546, suggest to name the vaccine studied for the cited literature for clarity.

Reviewer #2 (Remarks to the Author):

The authors have satisfactorily addressed all my concerns.

We thank this reviewer for the thorough review of our manuscript.

Reviewer #3 (Remarks to the Author):

Thank you for your detailed response to all of the Reviewer comments. Thank you for reinforcing in your comments that this manuscript is looking at the acute phase of CI-13 infection (day 4,8) due to the OMM12 mice losing weight that reaches the humane endpoint of the study. This was stated by the authors in the original and revised manuscript (Ln 126) but was missed/misinterpreted by this Reviewer (please make a spelling correction “human” to “humane” on Ln 126). This is a stark observation that 100% (or at least the majority) of OMM12 mice needed to be euthanized due to weight loss while 0% of the other groups did. It strengthens the important phenotypic difference observed between the SPF /GF and OMM12 mice. Please include a graph visually showing this data to the reader and detail what the humane endpoint criteria was for your study. Similar to a survival graph but with time when the humane endpoint was reached. This information is important to appreciate the extent of disease pathology reached with the OMM12 mice and also valuable data for future researchers to have to provide rationale to adjust their humane endpoint criteria as needed.

We have corrected “human” to “humane”.

We want to clarify that the humane endpoint is defined by animal welfare criteria that is reached, among others, when the body weight loss surpasses 20%. This prevented us from performing experiments beyond day 8 of infection. As one can see in Figure 1B, 2B and 3B, all animals lost body weight but OMM12 animals lost substantially more body weight compared to SPF animals seen already at day 4. Nevertheless and as also seen in other studies, there is considerable variability and also some of the germfree animals reached humane endpoint as defined by body weight loss only at day 8 post infection. Consequently, all animals were analyzed at day 8 post infection. Given that we show this parameter (body weight loss) already, we feel an additional graph with the same parameter would not add any new information. We have solved this problem by including a dotted line at 20% body weight loss to indicate humane endpoint and have now also included this information in the figure legend.

1. Thank you for the Authors response to this point and included several references on the impact of CD4 depletion in LCMV CI-13 infection. We agree these studies focus on the loss of function of CD8 T cells at chronic timepoints after infection but this manuscript could not investigate this later timepoint due to the OMM12 mice reaching humane endpoint.

Thank you for acknowledging this.

The viral titers in Figure 2C demonstrate that CD4 depleted OMM12 mice do not observe an increase in viral titers compared to OMM12 mice with CD4 T cells. Since the goal of the experiment in Figure 2 was to assess if the microbiome (OMM12) acted through CD4 T cells to alter disease pathology and viral clearance (Ln 157-158), the authors should state in this section that while the CD4 T cells are driving weight loss (disease pathology) in OMM12 CI-13 infected mice, CD4 T cells are not the causative agent that drive diminished viral titers in OMM12 mice. Thank you for pointing this out. Indeed, CD4 T cells do not seem to contribute to viral clearance but only to body weight loss. This is now better indicated in this section (Ln 176-77 and Ln 192) and not stated anymore in the discussion (Ln 399).

2. Thank you for the Authors response to this point and highlighting again that OMM12 mice reach a humane endpoint early and thus the humoral response later in the infection can not be assessed.

Thank you. Yes, this would indeed require another (milder) infection model.

3. Thank you for adjusting the FACS plots in Figure 1F to show IFN γ /TNF α . A point of clarity,

Zentrum für Allergie und Umwelt
Center of Allergy Research Munich

HELMHOLTZ
MUNICH

this Reviewer was asking for the total number of IFN γ /TNF α double positive cells from Fig 1F to be graphed out. Similar to the quantification of absolute number of Th1 CD4 cells in Fig 1E.

Unfortunately, we don't have total cell numbers from all experiments and therefore refrained from including total cell numbers. As a matter of fact, efficiency of cell isolation from the lamina propria often varies according to our experience (in contrast to lymphoid organs) and we believe that frequencies often give a better picture at this site compared to total cell numbers (which often rather reflect cell isolation efficiency).

We thank all the reviewers for their time and thorough review of our manuscript that substantially helped to improve and strengthen the clarity of our manuscript.

PD Dr. Caspar Ohnmacht

ZAUM
Center of Allergy & Environment
Biedersteiner Str. 29
80802 Munich
Germany

T + 49 (0) 89 41 40 34 51
F + 49 (0) 89 41 40 34 52

<http://www.zaum-online.de>
zaum@tum.de

Pt-to-Pt reply revised manuscript

Date: 19th of March 2025

Name of corresponding author: Caspar Ohnmacht

Title: A specific microbial consortium enhances Th1 immunity, improves LCMV viral clearance but aggravates LCMV disease pathology in mice

Manuscript number: NCOMMS-24-08228B

REVIEWER COMMENTS

A detailed color-coded pt-to-pt reply is provided below.

Reviewer #1 (Remarks to the Author):

The authors have satisfactorily addressed all my concerns. My thanks for all their detailed answers.

Thank you for the revision of our manuscript.

Reviewer #3 (Remarks to the Author):

In reference to Point 1:

Please include the humane endpoint criteria as defined by animal welfare criteria that was used in this study. Future studies and investigator will great benefit from this data as rationale to seek alternative metrics for a humane endpoint.

We have included a respective statement in the Methods section.

In reference to Point 3:

While efficiency of cell recovery from lamina propria prep is a technical challenge, the authors were able to display absolute number of T-bet⁺ cells from the small intestine in Fig. 1E. It is inconsistent to display absolute numbers of cells in Fig 1E but not Fig 1G since both pieces of data required lamina propria isolation. This Reviewer suggests display the data from both as relative absolute number. For Fig 1E this would be Tbet⁺ CD4 T cells per 1×10^6 CD45⁺ cells and Fig 1G as cytokine⁺ CD4 T cells per 1×10^6 CD45⁺ cells. These readouts are still relative numbers but takes into account relative abundance of parent CD4 T cell populations.

We thank the reviewer for this suggestion. There are specific reasons why we refrained from accepting this suggestion: First, Fig. 1E shows cells that were stained and fixed right after

cell isolation whereas as Fig. 1G depicts cells after restimulation. According to our experience, cell isolation from the intestinal tract followed by restimulation results in cell death of many non-T cells (e.g. granulocytes and other myeloid cells). Thus, calculating relative absolute numbers could still be misleading and we would most likely generate a systemic bias. Second, depicting frequencies of T cells (e.g. Tregs, Th1, Th2, Th17 cells, cytokine+ T cells etc.) is performed in most other studies because it is of general interest how the pool of T helper cells is composed and not so much absolute cell numbers. Third, presenting relative absolute cell numbers may be confounded by changes in the frequencies by other non-T cell populations.

Lastly, we could in principle offer to remove the plots of absolute Th1 cells from Fig. 1E for consistency with Fig. 1G but would prefer to keep it for the reasons depicted above.